# MERGETUNE: CONTINUED FINE-TUNING OF VISION-LANGUAGE MODELS

**Wenqing Wang[1]\*, Da Li[2,3]\*, Xiatian Zhu[1]†, Josef Kittler[1]†**

[1] University of Surrey
[2] Samsung AI Centre Cambridge
[3] Queen Mary University of London

https://github.com/Surrey-UP-Lab/MERGETUNE

## ABSTRACT

Fine-tuning vision-language models (VLMs) such as CLIP often leads to catastrophic forgetting of pretrained knowledge. Prior work primarily aims to mitigate forgetting during adaptation; however, forgetting often remains inevitable during this process. We introduce a novel paradigm, *continued fine-tuning (CFT)*, which seeks to recover pretrained knowledge after a zero-shot model has already been adapted. We propose a simple, model-agnostic CFT strategy (named MERGE-TUNE) guided by linear mode connectivity (LMC), which can be applied post hoc to existing fine-tuned models without requiring architectural changes. Given a fine-tuned model, we continue fine-tuning its trainable parameters (e.g., soft prompts or linear heads) to search for a continued model which has two low-loss paths to the zero-shot (e.g., CLIP) and the fine-tuned (e.g., CoOp) solutions. By exploiting the geometry of the loss landscape, the continued model implicitly merges the two solutions, restoring pretrained knowledge lost in the fine-tuned counterpart. A challenge is that the vanilla LMC constraint requires data replay from the pretraining task. We approximate this constraint for the zero-shot model via a second-order surrogate, eliminating the need for large-scale data replay. Experiments show that MERGETUNE improves the harmonic mean of CoOp by +5.6% on base-novel generalisation without adding parameters. On robust fine-tuning evaluations, the LMC-merged model from MERGETUNE surpasses ensemble baselines with lower inference cost, achieving further gains and state-of-the-art results when ensembled with the zero-shot model.

## 1 INTRODUCTION

Foundational vision-language models (VLMs) such as CLIP (Radford et al., 2021) have achieved strong zero-shot generalisation by pretraining on web-scale image-text pairs. To further adapt these models for downstream tasks, fine-tuning is often necessary. However, a well-known drawback of fine-tuning VLMs is catastrophic forgetting of pretrained knowledge, which weakens generalisation.

A range of fine-tuning strategies have been proposed to address this issue, though under different evaluation protocols. Parameter-efficient fine-tuning (PEFT) methods update only lightweight modules such as prompts (Zhou et al., 2022b;a; Yao et al., 2023; Li et al., 2025) or adapters (Yang et al., 2024; Khattak et al., 2023), achieving strong adaptation with relatively a smaller amount of trainable parameters. These approaches are primarily evaluated under few-shot settings, focusing on base-to-novel generalisation and, to a lesser extent, cross-domain and cross-dataset generalisation. In parallel, robust fine-tuning approaches (Wortsman et al., 2022b; Zhu et al., 2024) combine pretrained and finetuned models through ensembling, either in weight space (Wortsman et al., 2022a; Huang et al., 2024) or prediction space (Lakshminarayanan et al., 2017). Unlike PEFT methods, robust

---

\*Equal contribution
†Joint last authorship

fine-tuning methods are typically evaluated in many-shot settings, aligning with standard domain generalisation benchmarks (Zhou et al., 2023) while enforcing great in-distribution performance.

While PEFT methods mitigate forgetting by restricting updates to lightweight modules, they not only rely on increasingly complex modules (Yang et al., 2024) to balance adaptation with generalisation, but also still preserve pretrained knowledge incompletely. For example, as shown in Figure 1, no single prior PEFT method consistently outperforms CLIP across all 11 evaluation datasets without leveraging external knowledge (Yao et al., 2024). Similarly, ensembling can ease forgetting in the robust fine-tuning, but they often fail to fully reconcile pretrained knowledge with downstream adaptation (Zhu et al., 2024) and yield unstable performance (as shown in Table 4).

In this paper, we explore how to recover forgotten knowledge after fine-tuning is completed, which resembles the purpose of model ensembling methods in the robust fine-tuning (Wortsman et al., 2022b; Zhu et al., 2024). However, unlike ensembling, we take it as a post hoc direction to the VLM adaptation techniques through *continued fine-tuning*.

We take inspiration from recent advanced weight-space ensembling – model merging (Yadav et al., 2023; Huang et al., 2024), which demonstrates that different models can be combined to integrate complementary knowledge. Building on these insights, we propose to merge a fine-tuned VLM with its zero-shot counterpart to restore lost knowledge and enhance performance. However, ex-

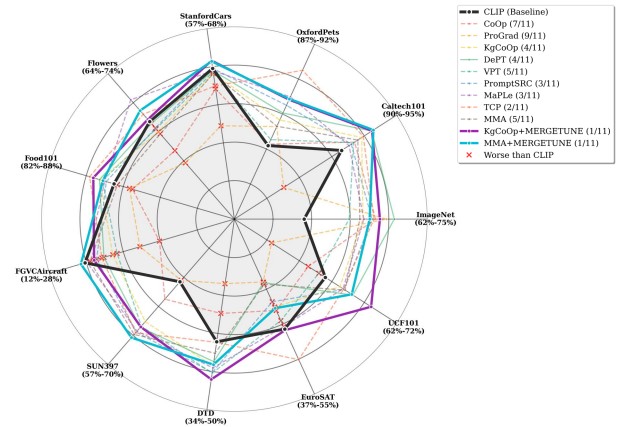

Figure 1: Cross-dataset generalisation shows no single PEFT method consistently outperforms CLIP across all 11 datasets, implying incomplete preservation of pretrained knowledge. Numbers in brackets (X/11) indicate X times a method underperforms CLIP.

isting model merging methods are often ineffective in this context, where the zero-shot and fine-tuned models could lie far apart in weight space. This weight-space gap could break the mode connectivity (Garipov et al., 2018) required for model merging to be effective and may explain the unstable performance commonly observed in weight averaging approaches (Wortsman et al., 2022a; Zhu et al., 2024). To overcome this, we propose a learning-based merging method that leverages linear mode connectivity (LMC) (Garipov et al., 2018) as an objective to integrate knowledge from both zero-shot and finetuned VLMs through continued fine-tuning.

Our model-agnostic MERGETUNE applies post hoc to already finetuned VLMs without requiring architectural changes. We can continue updating its trainable parameters (e.g., soft prompts, adapters, or linear heads) accordingly to search for weights that are connected to the zero-shot and the fine-tuned ones with two low-loss paths. The continued fine-tuning by searching these geometrically linear paths allows the continued model to effectively integrate pretrained knowledge into the fine-tuned model. However, the vanilla LMC constraint requires replaying the data used for training the zero-shot model (Mirzadeh et al., 2021), which is problematic. For example, the web-scale corpus used for pretraining CLIP is publicly inaccessible (Radford et al., 2021). And even accessible for replay, it is computationally prohibitive. We thus introduce a second-order surrogate that approximates the constraint for the zero-shot model, eliminating the need for large-scale data replay.

We apply MERGETUNE to multiple PEFT and robust fine-tuning methods. Experiments demonstrate that MERGETUNE consistently improves accuracies on the benchmarks for which these models were trained. On base-to-novel generalisation, MERGETUNE improves the harmonic mean of CoOp (Zhou et al., 2022b) by +5.6% without adding parameters. On robust fine-tuning tasks, the LMC-merged model from MERGETUNE achieves stronger out-of-distribution (OOD) generalisation than ensemble-based baselines, while maintaining comparable in-distribution (ID) accuracy. It also reduces inference cost and consistently outperforms CLIP across all evaluated cases. Moreover, simple ensembling with the zero-shot model brings further gains, setting new state-of-the-art results.

We summarise our contributions as follows: 1) While existing VLM methods seek to mitigate knowledge forgetting during downstream fine-tuning, we find that forgetting remains often unavoidable. In response, we propose *continued fine-tuning* — a new paradigm to existing methods — to restore forgotten knowledge after the model has been adapted. 2) Inspired by model merging (Section 3), which integrates complementary knowledge across models, we introduce a learning-based merging method MERGETUNE guided by linear mode connectivity (Section 4). 3) MERGETUNE is model-agnostic and can be applied post hoc to any fine-tuned VLM without architectural changes. Extensive experiments demonstrate its effectiveness in restoring pretrained knowledge to enhance the fine-tuned model (Section 5).

## 2 RELATED WORK

**VLM fine-tuning.**   Few-shot parameter-efficient fine-tuning (PEFT) has become a common strategy for adapting vision-language models (VLMs), as it reduces computational costs while mitigating catastrophic forgetting. Prompt learning methods, such as CoOp (Zhou et al., 2022b) and CoCoOp (Zhou et al., 2022a), optimise continuous prompts (task-specific or image-conditioned) instead of fixed templates, with extensions including prompt distributions (Lu et al., 2022) and attribute-guided prompts (Li et al., 2025). Adapter-based methods insert lightweight trainable modules, either uni-modal (Clip-Adapter (Gao et al., 2024), Tip-Adapter (Zhang et al., 2022)) or multi-modal (Yang et al., 2024; Jiang et al., 2025; Zang et al., 2022). Most of these existing methods focus on the few-shot learning evaluation of base-to-novel generalisation, with limited attention to cross-domain and cross-dataset generalisation.

Another line of work focuses on many-shot robust fine-tuning (Zhou et al., 2023), aiming to boost in-distribution accuracy without compromising out-of-distribution generalisation (Kumar et al., 2022; Wortsman et al., 2022b; Zhu et al., 2024). One common practice is weight ensembling after the model is fine-tuned, which blends zero-shot and fine-tuned model parameters to preserve both task-specific and general capabilities. Wortsman et al. (2022b) demonstrated that simple linear interpolation between zero-shot and fine-tuned weights can improve both ID and OOD performance. More sophisticated ensemble methods have emerged, such as VRF (Zhu et al., 2024), which conditionally combines models based on a "zero-shot failure" set, and R-Adapter (Kim et al., 2024), which integrates self-ensembling with lightweight adaptation. However, these complex ensembling procedures add inference overhead.

**Model merging.**   Model merging demonstrates that different models can be effectively combined to integrate complementary knowledge, grounded in the discovery of mode connectivity (Garipov et al., 2018). Early work showed that independently trained solutions can be linked by low-loss paths in weight space (Garipov et al., 2018; Draxler et al., 2018). Building on these insights, practical methods emerged: model soups combine multiple fine-tuned weights (Wortsman et al., 2022a), task arithmetic edits task vectors (Ilharco et al., 2023), and approaches like TIES-Merging (Yadav et al., 2023) and DARE (Yu et al., 2024) address interference during model merging.

Instead of merely mitigating forgetting during downstream fine-tuning, our goal is to restore the pretrained knowledge lost after the zero-shot model has been adapted. To this end, we introduce a *continued fine-tuning* strategy MERGETUNE that learns to merge the zero-shot and adapted models.

## 3 PRELIMINARIES

We begin by introducing model merging and mode connectivity, which underpin our approach.

**Model merging and mode connectivity.** When two models are trained on the same task with the loss $\mathcal{L}(\cdot)$ but differ in initialisation or training trajectory, they converge to different solutions $\hat{w}_1, \hat{w}_2$ that both achieve low loss. Often, one can merge them through weight averaging,

$$w = \gamma(\alpha) = (1 - \alpha)\hat{w}_1 + \alpha\hat{w}_2, \quad \alpha \in [0, 1]. \tag{1}$$

This linear interpolation can sometimes yield a model with performance comparable to its endpoints $\hat{w}_1$ and $\hat{w}_2$ (Izmailov et al., 2018) or yield an interesting observation (Garipov et al., 2018) as follows

$$\mathcal{L}(\gamma(\alpha)) \approx 0, \tag{2}$$

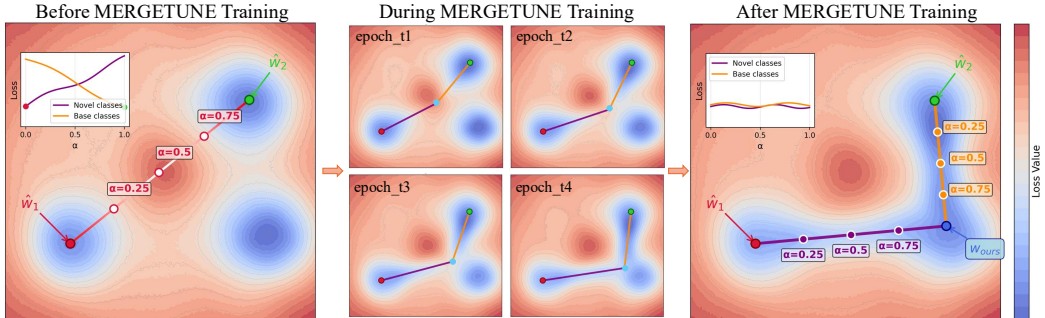

Figure 2: The proposed MERGETUNE (conceptual illustration). (Left) Before MERGETUNE Training: The zero-shot model $\hat{w}_1$ and fine-tuned $\hat{w}_2$ exist in separate minima with no low-loss connectivity. Linear interpolation between them (shown in the inset) reveals high barriers and induces a performance trade-off on base and novel classes. (Middle) During training, $w$ is searched to mode connected to both $\hat{w}_1$ and $\hat{w}_2$, gradually integrating both models. (Right) After MERGETUNE Training: Our continued model $w_{ours}$ merging two endpoints will be used for inference of both tasks $\hat{w}_1$ and $\hat{w}_2$ where trained. The two distinct low-loss paths, $\hat{w}_1 \rightarrow w_{ours}$ and $\hat{w}_2 \rightarrow w_{ours}$, show smooth interpolation curves (inset) indicating stable performance.

i.e. during the interpolation, the computed loss of the resulting model is consistently close to 0.

The effectiveness of model merging and the root of the observation in Eq. 1 can be explained by the phenomenon of mode connectivity (Garipov et al., 2018; Draxler et al., 2018), i.e. linear mode here. Empirical studies have shown that seemingly distinct optima discovered by independent training runs can be linked by continuous low-loss paths in parameter space. This indicates that neural network solutions are not totally isolated minima but could lie in connected valleys of the loss landscape. Mode connectivity thus provides a theoretical basis for why interpolation between models can preserve low loss.

**Merging models of different tasks.** Beyond the same-task setting above, model merging can be extended to integrate knowledge from different but related tasks, e.g., combining a model with another trained for different tasks (Mirzadeh et al., 2021; Yadav et al., 2023; Yu et al., 2024). In this case, the goal is to find a merged solution $w$ that preserves performance on both tasks. Formally, given two models $\hat{w}_1$ and $\hat{w}_2$ trained with task losses $\mathcal{L}_1$ and $\mathcal{L}_2$, one can seek a new model $w$ whose interpolation paths to $\hat{w}_1$ and $\hat{w}_2$

$$\gamma_1(\alpha) = \hat{w}_1 + \alpha(w - \hat{w}_1), \quad \gamma_2(\alpha) = \hat{w}_2 + \alpha(w - \hat{w}_2), \quad \alpha \in [0, 1], \tag{3}$$

satisfy

$$\mathcal{L}_1(\gamma_1(\alpha)) \approx 0, \quad \mathcal{L}_2(\gamma_2(\alpha)) \approx 0. \tag{4}$$

That is, throughout the interpolation, model $w$ maintains two smooth, low-loss connections to both endpoints, ensuring that knowledge from both tasks is preserved in the single $w$ (Mirzadeh et al., 2021; Yadav et al., 2023; Yu et al., 2024).

From this perspective, model merging can be viewed as finding a solution that lies on low-loss paths connecting multiple checkpoints. This geometric view unifies heuristic averaging methods, such as training-free approaches TIES (Yadav et al., 2023) and DARE (Yu et al., 2024), with regularisation-based methods (Mirzadeh et al., 2021). Enforcing mode connectivity ensures that the merged model integrates the endpoints, preserving knowledge from both. However, training-free methods (Yadav et al., 2023; Yu et al., 2024) lack explicit enforcement of mode connectivity, and may therefore struggle to integrate models effectively.

# 4 MERGETUNE: CONTINUED FINETUNING OF VLMS VIA MODE-CONNECTIVITY GUIDANCE

We now introduce our continued fine-tuning (CFT) objective, which enforces *linear mode connectivity* between the continued model and both the zero-shot and fine-tuned solutions. This ensures that the final model preserves pretrained knowledge while retaining downstream adaptation.

**Linear mode connectivity as an objective.** The effectiveness of model merging is grounded in the phenomenon of linear mode connectivity. We employ it directly as a learning principle, as inspired by Eq. 3 and Eq. 4. Our hypothesis is that if a model can be linearly connected to another solution through a consistently low-loss path, it can inherit and preserve the knowledge of that solution.

Concretely, given a zero-shot checkpoint $\hat{w}_1$ (e.g., CLIP) and a downstream fine-tuned checkpoint $\hat{w}_2$ (e.g., CoOp), we seek a continued solution $w$ that integrates both. We require $w$ to remain linearly connected to both $\hat{w}_1$ and $\hat{w}_2$ via low-loss interpolations. This leads to the objective:

$$w = \arg\min_w \mathbb{E}_{\alpha\sim\mathcal{U}[0,1]}\Big[\mathcal{L}_1(\hat{w}_1 + \alpha(w - \hat{w}_1)) + \mathcal{L}_2(\hat{w}_2 + \alpha(w - \hat{w}_2))\Big], \tag{5}$$

where $\mathcal{L}_1$ and $\mathcal{L}_2$ are the pretraining and downstream training losses for the zero-shot and fine-tuned models, and $\alpha$ is the interpolation coefficient uniformly sampled from $[0, 1]$. This formulation leverages linear mode connectivity as a learning objective to merge knowledge from two solutions. After training, $w$ is used for inference. An illustration of our training process is shown in Figure 2.

**Challenge.** A direct implementation of Eq. 5 requires the computation of both $\mathcal{L}_1$ and $\mathcal{L}_2$. However, $\mathcal{L}_1$ depends on the pretraining data (e.g., CLIP's web-scale corpus), which is often inaccessible and computationally prohibitive to replay. Thus, we propose a second-order surrogate loss to approximate the computation of Task 1 loss term in a replay-free manner.

**Second-order surrogate.** We approximate the Task 1 interpolation term using a second-order Taylor expansion:

$$\mathcal{L}_1(\hat{w}_1 + \alpha(w - \hat{w}_1)) \approx \mathcal{L}_1(\hat{w}_1) + \alpha\nabla\mathcal{L}_1(\hat{w}_1)^\top(w - \hat{w}_1) + \tfrac{\alpha^2}{2}(w - \hat{w}_1)^\top H_1(w - \hat{w}_1), \tag{6}$$

where $H_1 = \nabla^2\mathcal{L}_1(\hat{w}_1)$ is the Hessian matrix.

We adopt two assumptions: (1) $\nabla\mathcal{L}_1(\hat{w}_1) \approx 0$, as $\hat{w}_1$ lies near a local optimum for Task 1. (2) $H_1 \approx \mu I$, assuming isotropic curvature for tractability following common practices. Under these assumptions, Eq. 6 simplifies to:

$$\mathcal{L}_1(\hat{w}_1 + \alpha(w - \hat{w}_1)) \approx \mathcal{L}_1(\hat{w}_1) + \tfrac{\mu\alpha^2}{2}\|w - \hat{w}_1\|^2. \tag{7}$$

$\mathcal{L}_1(\hat{w}_1)$ is a constant value, and $\frac{\mu\alpha^2}{2}$ can be represented by $\lambda$. The surrogate regulairiser can be formulated as:

$$\mathcal{R}_{\text{Task1}} = \lambda\|w - \hat{w}_1\|^2. \tag{8}$$

**Final replay-free objective.** Combining the Task 1 surrogate with the Task 2 objectives leads to our final continued fine-tuning loss:

$$\mathcal{L}(w) = \mathcal{L}_2(w) + \lambda\|w - \hat{w}_1\|^2 + \beta\,\mathbb{E}_{\alpha\sim\mathcal{U}[0,1)}\mathcal{L}_2(\hat{w}_2 + \alpha(w - \hat{w}_2))\,. \tag{9}$$

This objective preserves pretrained knowledge through proximity to $\hat{w}_1$, while enforcing low-loss connectivity to the downstream solution $\hat{w}_2$. Empirically, we decoupled $\mathcal{L}_2(w)$, where $\alpha = 1$, from the expectation term. In this way, the LMC terms serve as a regulariser for training. Also, the expectation in the third term is computationally intractable. We thus approximate it by evaluating a small number of evenly spaced $\alpha$ values (e.g., five or ten), which yields promising results.

**Application to CLIP fine-tuning methods.** Our framework makes no limitation on which parameters are trainable, allowing it to be applied broadly. We optimise $w$ under the same training configurations as the base method: In prompt-based methods such as CoOp and KgCoOp[1], $w = T(p)$ corresponds to the classifier weights derived from learnable prompts $p$ and $T(\cdot)$ is the text encoder. In adapter-based methods such as MMA, $w = T(p; \theta)$ includes fixed prompts and trainable multimodal adapters $\theta$. In many-shot regimes, $w$ can represent the linear classification head (linear probing) or the entire model parameters (end-to-end fine-tuning). This flexibility allows our continued fine-tuning approach to act as a general post hoc enhancement, seamlessly integrating with diverse CLIP adaptation strategies.

Table 1: Base-to-novel generalisation experiments on 11 datasets. Our method achieves consistent average performance improvement over different baselines. †: Using large language model or teacher model's knowledge.

| Method | Average | | | ImageNet | | | Caltech101 | | | OxfordPets | | |
|---|---|---|---|---|---|---|---|---|---|---|---|---|
| | Base | Novel | HM | Base | Novel | HM | Base | Novel | HM | Base | Novel | HM |
| CLIP (ICML 21) | 69.34 | 74.22 | 71.70 | 72.43 | 68.14 | 70.22 | 96.84 | 94.00 | 95.40 | 91.17 | 97.26 | 94.12 |
| CoOp (IJCV 22) | 82.69 | 63.22 | 71.66 | 76.47 | 67.88 | 71.92 | 98.00 | 89.81 | 93.73 | 93.67 | 95.29 | 94.47 |
| KgCoOp (CVPR 22) | 80.73 | 73.61 | 77.01 | 75.83 | 69.96 | 72.78 | 97.72 | 94.39 | 96.03 | 94.65 | 97.76 | 96.18 |
| MaPLe (CVPR 23) | 82.28 | 75.14 | 78.55 | 76.66 | 70.54 | 73.47 | 97.74 | 94.36 | 96.02 | 95.43 | 97.76 | 96.58 |
| PromptSRC (ICCV 23) | 84.26 | 76.10 | 79.97 | 77.60 | 70.73 | 74.01 | 98.10 | 94.03 | 96.02 | 95.33 | 97.30 | 96.30 |
| MMA (CVPR 24) | 83.20 | 76.80 | 79.87 | 77.31 | 71.00 | 74.02 | 98.40 | 94.00 | 96.15 | 95.40 | 98.07 | 96.72 |
| CoPrompt† (ICLR 24) | 84.00 | 77.23 | 80.48 | 77.67 | 71.27 | 74.33 | 98.27 | 94.90 | 96.55 | 95.67 | 98.10 | 96.87 |
| PromptKD† (CVPR 24) | 86.96 | 80.73 | 83.73 | 80.83 | 74.66 | 77.62 | 98.91 | 96.65 | 97.77 | 96.30 | 98.01 | 97.15 |
| CoOp + ATPrompt† (ICCV 25) | 82.68 | 68.04 | 74.65 (+2.99) | 76.27 | 70.60 | 73.33 | 97.95 | 93.63 | 95.74 | 94.77 | 96.59 | 95.67 |
| PromptKD + ATPrompt† (ICCV 25) | 87.05 | 81.82 | 84.35 (+0.62) | 80.90 | 74.83 | 77.75 | 98.90 | 96.52 | 97.70 | 96.92 | 98.27 | 97.59 |
| CoOp + TIES (NeurIPS 23) | 66.95 | 65.71 | 66.32 (-5.33) | 70.09 | 61.65 | 65.60 | 93.63 | 91.34 | 92.50 | 90.12 | 95.13 | 92.56 |
| CoOp + DARE (ICML 24) | 75.91 | 65.97 | 70.59 (-1.07) | 74.25 | 64.32 | 68.93 | 96.36 | 91.12 | 93.67 | 93.41 | 96.33 | 94.84 |
| CoOp + MERGETUNE | 80.82 | 73.97 | 77.24 (+5.58) | 75.96 | 69.91 | 72.81 | 97.91 | 94.65 | 96.25 | 95.09 | 97.75 | 96.40 |
| KgCoOp + TIES (NeurIPS 23) | 73.03 | 72.09 | 72.56 (-4.45) | 74.06 | 68.06 | 70.93 | 97.27 | 94.43 | 95.83 | 93.99 | 96.51 | 95.23 |
| KgCoOp + DARE (ICML 24) | 78.15 | 72.41 | 75.17 (-1.84) | 75.10 | 68.73 | 71.78 | 97.53 | 94.47 | 95.97 | 92.22 | 97.37 | 94.73 |
| KgCoOp + MERGETUNE | 81.85 | 74.46 | 77.98 (+0.97) | 76.49 | 69.35 | 72.75 | 97.87 | 94.91 | 96.37 | 95.32 | 97.76 | 96.53 |
| MMA + TIES (NeurIPS 23) | 70.39 | 68.46 | 69.41 (-10.46) | 73.16 | 66.48 | 69.66 | 95.36 | 92.23 | 93.77 | 91.24 | 93.11 | 92.17 |
| MMA + DARE (ICML 24) | 74.25 | 69.52 | 71.81 (-8.06) | 74.17 | 66.76 | 70.27 | 94.95 | 92.49 | 93.70 | 90.52 | 95.53 | 92.96 |
| MMA + MERGETUNE | 84.27 | 76.94 | 80.44 (+0.57) | 77.68 | 70.65 | 74.00 | 98.32 | 94.65 | 96.45 | 95.75 | 98.10 | 96.91 |
| PromptKD† + TIES (NeurIPS 23) | 82.03 | 77.16 | 79.52 (-4.21) | 76.12 | 69.32 | 72.56 | 97.61 | 94.9 | 96.24 | 95.23 | 96.13 | 95.68 |
| PromptKD† + DARE (ICML 24) | 84.76 | 79.65 | 82.13 (-1.6) | 78.29 | 72.54 | 75.31 | 98.12 | 96.19 | 97.15 | 95.56 | 98.00 | 96.76 |
| PromptKD† + MERGETUNE | 87.23 | 81.17 | 84.09 (+0.36) | 80.89 | 74.88 | 77.77 | 98.93 | 96.64 | 97.77 | 96.63 | 98.34 | 97.48 |

| Method | StanfordCars | | | Flowers102 | | | Food101 | | | FGVCAircraft | | |
|---|---|---|---|---|---|---|---|---|---|---|---|---|
| | Base | Novel | HM | Base | Novel | HM | Base | Novel | HM | Base | Novel | HM |
| CLIP (ICML 21) | 63.37 | 74.89 | 68.65 | 72.08 | 77.80 | 74.83 | 90.10 | 91.22 | 90.66 | 27.19 | 36.29 | 31.09 |
| CoOp (IJCV 22) | 78.12 | 60.40 | 68.13 | 97.60 | 59.67 | 74.06 | 88.33 | 82.26 | 85.19 | 40.44 | 22.30 | 28.75 |
| KgCoOp (CVPR 22) | 71.76 | 75.04 | 73.36 | 95.00 | 74.73 | 83.65 | 90.50 | 91.70 | 91.10 | 36.21 | 33.55 | 34.83 |
| MaPLe (CVPR 23) | 72.94 | 74.00 | 73.47 | 95.92 | 72.46 | 82.56 | 90.71 | 92.05 | 91.38 | 37.44 | 35.61 | 36.50 |
| PromptSRC (ICCV 23) | 78.27 | 74.97 | 76.58 | 98.07 | 76.50 | 85.95 | 90.67 | 91.53 | 91.10 | 42.73 | 37.87 | 40.15 |
| MMA (CVPR 24) | 78.50 | 73.10 | 75.70 | 97.77 | 75.93 | 85.48 | 90.13 | 91.30 | 90.71 | 40.57 | 36.33 | 38.33 |
| CoPrompt† (ICLR 24) | 76.97 | 74.40 | 75.66 | 97.27 | 76.60 | 85.71 | 90.73 | 92.07 | 91.40 | 40.20 | 39.33 | 39.76 |
| PromptKD† (CVPR 24) | 82.80 | 83.37 | 83.13 | 99.42 | 82.62 | 90.24 | 92.43 | 93.68 | 93.05 | 49.12 | 41.81 | 45.17 |
| CoOp + ATPrompt† (ICCV 25) | 77.43 | 66.55 | 71.58 | 97.44 | 67.52 | 79.77 | 90.77 | 87.44 | 88.09 | 40.38 | 27.22 | 32.52 |
| PromptKD + ATPrompt† (ICCV 25) | 82.51 | 84.03 | 83.26 | 99.15 | 82.03 | 89.78 | 92.48 | 93.86 | 93.22 | 49.63 | 42.35 | 45.70 |
| CoOp + TIES (NeurIPS 23) | 57.94 | 63.42 | 60.56 | 73.60 | 65.41 | 69.27 | 88.42 | 87.64 | 88.02 | 30.29 | 28.83 | 29.54 |
| CoOp + DARE (ICML 24) | 67.82 | 61.62 | 64.57 | 90.88 | 63.73 | 74.92 | 87.78 | 87.22 | 87.50 | 34.41 | 28.25 | 31.03 |
| CoOp + MERGETUNE | 71.94 | 75.12 | 73.49 | 95.32 | 74.30 | 83.51 | 90.56 | 91.75 | 91.15 | 36.17 | 34.35 | 35.24 |
| KgCoOp + TIES (NeurIPS 23) | 65.79 | 74.17 | 69.73 | 74.26 | 72.53 | 73.39 | 90.25 | 91.58 | 90.91 | 30.81 | 28.97 | 29.86 |
| KgCoOp + DARE (ICML 24) | 67.84 | 73.02 | 70.33 | 87.02 | 72.88 | 79.33 | 90.44 | 91.60 | 91.02 | 33.49 | 30.37 | 31.85 |
| KgCoOp + MERGETUNE | 73.21 | 75.14 | 74.17 | 96.49 | 74.68 | 84.20 | 90.51 | 91.83 | 91.17 | 37.09 | 35.11 | 36.07 |
| MMA + TIES (NeurIPS 23) | 60.72 | 70.12 | 65.08 | 73.46 | 70.74 | 72.07 | 88.79 | 89.83 | 89.31 | 29.27 | 28.06 | 28.65 |
| MMA + DARE (ICML 24) | 64.82 | 71.52 | 68.01 | 77.15 | 70.95 | 73.92 | 89.85 | 87.61 | 88.72 | 30.35 | 28.23 | 29.25 |
| MMA + MERGETUNE | 81.24 | 72.30 | 76.51 | 98.04 | 76.33 | 85.83 | 90.77 | 91.57 | 91.17 | 42.86 | 36.01 | 39.14 |
| PromptKD† + TIES (NeurIPS 23) | 75.13 | 76.99 | 76.05 | 94.53 | 79.22 | 86.20 | 90.96 | 92.10 | 91.53 | 43.68 | 38.75 | 41.07 |
| PromptKD† + DARE (ICML 24) | 80.13 | 81.95 | 81.03 | 97.73 | 80.93 | 88.54 | 91.57 | 93.28 | 92.42 | 46.96 | 40.79 | 43.66 |
| PromptKD† + MERGETUNE | 82.98 | 83.90 | 83.44 | 99.30 | 82.67 | 90.23 | 92.45 | 93.91 | 93.17 | 49.82 | 42.31 | 45.76 |

| Method | SUN397 | | | DTD | | | EuroSAT | | | UCF101 | | |
|---|---|---|---|---|---|---|---|---|---|---|---|---|
| | Base | Novel | HM | Base | Novel | HM | Base | Novel | HM | Base | Novel | HM |
| CLIP (ICML 21) | 69.36 | 75.35 | 72.23 | 53.24 | 59.90 | 56.37 | 56.48 | 64.05 | 60.03 | 70.53 | 77.50 | 73.85 |
| CoOp (IJCV 22) | 80.60 | 65.89 | 72.51 | 79.44 | 41.18 | 54.24 | 92.19 | 54.74 | 68.69 | 84.69 | 56.05 | 67.46 |
| CoCoOp (CVPR 22) | 79.74 | 76.86 | 78.27 | 77.01 | 56.00 | 64.85 | 87.49 | 60.04 | 71.21 | 82.33 | 73.45 | 77.64 |
| KgCoOp (CVPR 22) | 80.29 | 76.53 | 78.36 | 77.55 | 54.99 | 64.35 | 85.64 | 64.34 | 73.48 | 82.89 | 76.67 | 79.66 |
| MaPLe (CVPR 23) | 80.82 | 78.70 | 79.75 | 80.36 | 59.18 | 68.16 | 94.07 | 73.23 | 82.35 | 83.00 | 78.66 | 80.77 |
| PromptSRC (ICCV 23) | 82.67 | 78.47 | 80.52 | 83.37 | 62.97 | 71.75 | 92.90 | 73.90 | 82.32 | 87.10 | 78.80 | 82.74 |
| MMA (CVPR 24) | 82.27 | 78.57 | 80.38 | 83.20 | 65.63 | 73.38 | 85.46 | 82.34 | 83.87 | 86.23 | 80.03 | 82.20 |
| CoPrompt† (ICLR 24) | 82.63 | 80.03 | 81.30 | 83.13 | 64.73 | 72.79 | 94.60 | 78.57 | 85.84 | 86.90 | 79.57 | 83.07 |
| PromptKD† (CVPR 24) | 83.69 | 81.54 | 82.60 | 85.84 | 71.37 | 77.94 | 97.54 | 82.08 | 89.14 | 89.71 | 82.27 | 86.10 |
| CoOp + ATPrompt† (ICCV 25) | 80.84 | 68.64 | 74.24 | 80.83 | 45.49 | 58.22 | 90.34 | 59.79 | 71.96 | 84.49 | 64.96 | 73.45 |
| PromptKD + ATPrompt† (ICCV 25) | 83.87 | 81.35 | 82.59 | 86.92 | 72.34 | 78.96 | 97.05 | 92.07 | 94.49 | 89.29 | 82.44 | 85.73 |
| CoOp + TIES (NeurIPS 23) | 62.41 | 62.99 | 62.70 | 57.33 | 41.87 | 48.40 | 50.17 | 69.85 | 58.40 | 62.45 | 54.68 | 58.30 |
| CoOp + DARE (ICML 24) | 70.33 | 63.01 | 66.47 | 72.49 | 47.02 | 57.04 | 69.71 | 65.22 | 67.39 | 77.61 | 57.84 | 66.28 |
| CoOp + MERGETUNE | 80.37 | 76.19 | 78.22 | 78.51 | 56.84 | 65.93 | 85.15 | 64.51 | 73.40 | 82.04 | 78.33 | 80.14 |
| KgCoOp + TIES (NeurIPS 23) | 73.05 | 73.88 | 73.46 | 60.77 | 53.54 | 56.93 | 68.66 | 65.45 | 67.02 | 74.46 | 73.91 | 74.18 |
| KgCoOp + DARE (ICML 24) | 77.47 | 74.81 | 76.12 | 72.46 | 54.19 | 62.01 | 84.48 | 63.03 | 72.19 | 81.63 | 75.99 | 78.71 |
| KgCoOp + MERGETUNE | 80.53 | 77.28 | 78.87 | 79.90 | 57.33 | 66.75 | 89.56 | 67.89 | 77.22 | 83.35 | 77.76 | 80.45 |
| MMA + TIES (NeurIPS 23) | 69.46 | 65.77 | 67.56 | 55.38 | 48.36 | 51.63 | 65.69 | 67.95 | 66.80 | 71.74 | 60.42 | 65.60 |
| MMA + DARE (ICML 24) | 71.43 | 72.61 | 72.02 | 67.65 | 50.37 | 57.74 | 77.42 | 64.16 | 70.17 | 78.46 | 64.47 | 70.78 |
| MMA + MERGETUNE | 82.46 | 78.48 | 80.42 | 83.37 | 65.34 | 73.26 | 90.09 | 81.91 | 85.81 | 86.37 | 81.02 | 83.61 |
| PromptKD† + TIES (NeurIPS 23) | 79.11 | 76.49 | 77.78 | 78.86 | 65.93 | 71.82 | 88.64 | 81.59 | 84.97 | 82.47 | 77.32 | 79.81 |
| PromptKD† + DARE (ICML 24) | 80.34 | 79.66 | 80.00 | 84.22 | 70.04 | 76.48 | 92.48 | 82.11 | 86.99 | 86.97 | 80.68 | 83.71 |
| PromptKD† + MERGETUNE | 83.92 | 81.49 | 82.69 | 86.77 | 72.10 | 78.76 | 97.81 | 83.82 | 90.28 | 89.99 | 82.79 | 86.24 |

Table 2: Cross-dataset generalisation results. Models are trained on ImageNet and directly evaluated on other datasets. Avg-C = average over all cross-dataset targets. *: Our reproduction. †: Using external knowledge.

| Method | Caltech101 | Oxford Pets | Stanford Cars | Flowers102 | Food101 | FGVC Aircraft | SUN397 | DTD | EuroSAT | UCF101 | Avg-C |
|---|---|---|---|---|---|---|---|---|---|---|---|
| CLIP (ICML 21) | 93.30 | 89.10 | 65.70 | 70.70 | 85.90 | 24.90 | 62.60 | 44.30 | 48.30 | 67.60 | 65.24 |
| CoOp (IJCV 22) | 93.70 | 89.14 | 64.51 | 68.71 | 85.30 | 18.47 | 64.15 | 41.92 | 46.39 | 66.55 | 63.88 |
| +TIES (NeurIPS 23) | 93.31 | 88.25 | 65.51 | 67.40 | 85.23 | 23.91 | 62.56 | 44.39 | 42.22 | 65.24 | 63.80 (-0.08) |
| +DARE (ICML 24) | 89.67 | 86.40 | 62.87 | 66.32 | 84.27 | 18.07 | 60.27 | 38.55 | 45.65 | 64.68 | 61.67 (-2.21) |
| +MERGETUNE | 93.96 | 89.97 | 65.67 | 70.40 | 86.43 | 23.88 | 66.35 | 46.53 | 46.12 | 68.71 | 65.80 (+1.92) |
| KgCoOp (CVPR 22) | 93.92 | 89.83 | 65.41 | 70.01 | 86.36 | 22.51 | 66.16 | 46.35 | 46.04 | 68.50 | 65.51 |
| +TIES (NeurIPS 23) | 91.94 | 87.56 | 64.33 | 66.92 | 85.53 | 21.69 | 63.66 | 41.75 | 46.36 | 65.74 | 63.55 (-1.96) |
| +DARE (ICML 24) | 93.66 | 88.15 | 64.81 | 69.29 | 86.16 | 22.24 | 64.64 | 45.09 | 45.38 | 66.06 | 64.55 (-0.96) |
| +MERGETUNE | 94.24 | 90.41 | 66.15 | 70.85 | 86.57 | 24.12 | 66.63 | 47.46 | 48.41 | 70.42 | 66.53 (+1.02) |
| MMA* (CVPR 24) | 93.80 | 89.65 | 65.41 | 69.86 | 85.79 | 24.47 | 67.31 | 45.26 | 43.70 | 68.82 | 65.41 |
| +TIES (NeurIPS 23) | 90.54 | 86.21 | 62.46 | 65.35 | 84.21 | 17.98 | 60.33 | 38.35 | 41.69 | 64.13 | 61.13 (-4.28) |
| +DARE (ICML 24) | 91.24 | 87.14 | 63.25 | 66.45 | 84.43 | 19.46 | 62.12 | 40.44 | 42.89 | 65.35 | 62.28 (-3.13) |
| +MERGETUNE | 94.28 | 90.43 | 66.11 | 71.46 | 86.26 | 25.26 | 67.59 | 46.22 | 46.16 | 69.23 | 66.30 (+0.89) |
| PromptKD† (ICLR 24) | 93.61 | 91.59 | 73.93 | 75.33 | 88.84 | 26.24 | 68.57 | 55.08 | 63.74 | 76.39 | 71.33 |
| +TIES (NeurIPS 23) | 91.35 | 89.25 | 70.35 | 69.36 | 86.26 | 23.74 | 64.75 | 50.36 | 60.68 | 72.54 | 67.86 (-3.47) |
| +DARE (ICML 24) | 93.26 | 90.36 | 73.26 | 74.26 | 88.36 | 25.14 | 67.47 | 54.26 | 63.03 | 75.19 | 70.46 (-0.87) |
| +MERGETUNE | 93.95 | 92.01 | 74.17 | 75.39 | 89.32 | 26.58 | 68.61 | 55.74 | 63.94 | 77.12 | 71.68 (+0.35) |

## 5 EXPERIMENTS

We evaluate MERGETUNE across multiple benchmarks and protocols:

**Few-shot setting.** *Base-to-novel generalisation:* We evaluate on 11 datasets, each dataset is split into base and novel classes; models are trained on base classes and evaluated on both base and novel categories, with harmonic mean (HM) reported as the main metric. *Cross-dataset generalisation:* Following (Zhou et al., 2022a), models are trained on ImageNet (16 shots per class) and directly evaluated on other datasets without further adaptation. *Domain generalisation:* To assess robustness to domain shift, it is to evaluate ImageNet-finetuned models on four ImageNet variants.

**Many-shot setting.** *ID-OOD generalisation:* In the robust learning setting, models are initialized from pre-trained CLIP with an additional linear classifier head derived from CLIP's zero-shot text embeddings generated by encoding class-specific text prompts through the pre-trained text encoder, then fine-tuned on ImageNet-1K using standard supervised cross-entropy loss. The finetuned model is tested without any additional training or adaptation across both in-distribution and out-of-distribution settings. We report performance on the four domain-shifted ImageNet variants plus ObjectNet (Barbu et al., 2019). All reported results are averaged over three random seeds.

### 5.1 IMPLEMENTATION DETAILS

**Base models.** We evaluate our continued fine-tuning strategy on multiple VLM adaptation methods. For few-shot learning, we build upon CoOp (Zhou et al., 2022b), KgCoOp (Yao et al., 2023), MMA (Yang et al., 2024), and PromptKD (Li et al., 2024), while for many-shot settings we consider linear-head and full-model fine-tuning. For each baseline, we reproduce it to obtain a downstream checkpoint. We then apply MERGETUNE to this checkpoint together with the zero-shot CLIP model, following the same training configurations as the respective baseline. All experiments use CLIP ViT-B/16 backbone. Details on datasets, training, and evaluation are in the supplement.

**Competitors.** We compare our MERGETUNE against two families of competing methods. First, we compare with existing training-free model merging techniques, including TIES (Yadav et al., 2023), which mitigates parameter interference via task-wise sparsification; and DARE (Yu et al., 2024), which improves cross-task transfer through reweighting strategies. Second, we evaluate against representative ensembling approaches: Wise-FT (Wortsman et al., 2022b), which linearly interpolates pretrained and finetuned weights, and VRF (Zhu et al., 2024), which enhances robustness through

---

[1]Also interesting to note, our MERGETUNE reduces to KgCoOp when $\beta=0$ for prompt tuning.

variance-reduction ensembles. These methods provide strong baselines for assessing the effectiveness of our MERGETUNE.

## 5.2 FEW-SHOT BASE-TO-NOVEL GENERALISATION

Table 1 reports base-to-novel generalisation across 11 datasets. We extend CoOp, KgCoOp, and PromptKD with training-free model merging methods, including TIES, DARE, and our MERGE-TUNE. Since training-free merging cannot directly handle models with structural differences (e.g., MMA, whose adapted models differ from pretrained CLIP), we instead merge only the linear heads while preserving the original MMA image encoder.

The results highlight two key observations. First, merging zero-shot and fine-tuned checkpoints via TIES or DARE degrades performance. For example, CoOp+TIES reduces the HM by 5.33% compared to vanilla CoOp, while CoOp+DARE decreases it by 1.07%. Even when applied to stronger baselines such as PromptKD, these training-free approaches fail to improve performance, underscoring the difficulty of directly interpolating between CLIP and fine-tuned checkpoints.

Second, our MERGETUNE consistently improves generalisation across all baselines. The magnitude of improvement correlates inversely with baseline performance, which is theoretically consistent with our framework. Methods that already preserve pretrained knowledge well (e.g., PromptKD at 83.13% HM) have less forgotten knowledge to recover, thus showing smaller but consistent gains (+0.36%). Conversely, methods with more catastrophic forgetting (e.g., CoOp at 71.66% HM) benefit more substantially (+5.58%), as our LMC-based approach has more pretrained knowledge to restore. This demonstrates that MERGETUNE's effectiveness scales with the degree of forgetting, making it particularly valuable for improving methods that struggle with knowledge retention.

## 5.3 FEW-SHOT CROSS-DATASET GENERALISATION

Table 2 presents cross-dataset generalisation results, where models trained on ImageNet are directly evaluated on 10 other datasets without adaptation. Training-free merging methods (TIES and DARE) generally degrade performance, whereas MERGETUNE delivers consistent gains across all baselines, with average HM improvements of +1.92 on CoOp, +0.86 on KgCoOp, +0.89 on MMA, and +0.35 on PromptKD. The improvements are especially pronounced on challenging datasets such as FGVC Aircraft, DTD and EuroSAT. Notably, by merging pretrained knowledge from the zero-shot model via our MERGETUNE, MMA is able to surpass CLIP on all evaluated datasets.

## 5.4 FEW-SHOT DOMAIN GENERALISATION

We evaluate the models few-shot-tuned from ImageNet on various out-of-domain datasets shown in Table 3. Again, training-free merging approaches (TIES and DARE) gain negative performance gains across all baselines, with notable degradation on datasets featuring larger shifts, such as ImageNet-Sketch and ImageNet-Adversarial. Our MERGETUNE consistently outperforms all baselines against domain shifts, achieving positive gains of +0.87 for CoOp, +0.35 for KgCoOp, +0.42 for MMA, and +0.30 for PromptKD (as Avg-D), confirming enhanced robustness under distribution shift scenarios.

Table 3: Domain generalisation results on ImageNet and four distribution shifts. Avg-D = average over domain-shifted datasets. MMA* is our reproduction.

| Method | ImageNet | -V2 | -S | -A | -R | Avg-D |
|---|---|---|---|---|---|---|
| CoOp (IJCV 22) | 71.51 | 64.20 | 47.99 | 49.71 | 75.21 | 59.28 |
| +TIES (NeurIPS 23) | 62.84 | 56.26 | 41.14 | 44.65 | 70.77 | 53.20 (-6.08) |
| +DARE (ICML 24) | 69.02 | 61.75 | 45.87 | 49.10 | 73.85 | 57.64 (-1.64) |
| +MERGETUNE | 71.68 | 64.56 | 48.67 | 50.74 | 76.61 | 60.15 (+0.87) |
| KgCoOp (CVPR 22) | 70.66 | 64.10 | 48.97 | 50.69 | 76.70 | 60.11 |
| +TIES (NeurIPS 23) | 67.77 | 61.47 | 46.36 | 49.86 | 75.91 | 58.40 (-1.71) |
| +DARE (ICML 24) | 69.67 | 62.89 | 47.69 | 50.33 | 76.15 | 59.27 (-0.84) |
| +MERGETUNE | 71.80 | 64.70 | 49.10 | 51.01 | 77.02 | 60.46 (+0.35) |
| MMA* (CVPR 24) | 70.45 | 63.87 | 48.84 | 49.91 | 77.29 | 59.98 |
| +TIES (NeurIPS 23) | 64.01 | 57.13 | 42.32 | 45.23 | 72.11 | 54.20 (-5.78) |
| +DARE (ICML 24) | 68.35 | 58.67 | 44.75 | 47.26 | 74.26 | 56.24 (-3.74) |
| +MERGETUNE | 71.11 | 64.41 | 49.26 | 50.43 | 77.51 | 60.40 (+0.42) |
| PromptKD[†] (ICLR 24) | 77.12 | 69.77 | 58.72 | 70.36 | 87.01 | 71.47 |
| +TIES (NeurIPS 23) | 74.63 | 64.22 | 54.85 | 67.47 | 83.85 | 67.60 (-3.87) |
| +DARE (ICML 24) | 76.14 | 68.94 | 56.89 | 69.36 | 86.67 | 70.47 (-1.00) |
| +MERGETUNE | 77.14 | 70.21 | 58.97 | 70.68 | 87.22 | 71.77 (+0.30) |

Table 4: ID-OOD generaisation accuracy of various methods on ImageNet and distribution shifts for CLIP ViT-B/16 in the robust fine-tuning evaluation. Avg-D = average over domain-shifted datasets.

| Method | Imagenet | Distribution shifts | | | | ObjectNet | Avg-D |
|---|---|---|---|---|---|---|---|
| | | -V2 | -S | -A | -R | | |
| Zero-shot (CLIP) | 68.34 | 61.90 | 48.27 | 50.12 | 77.60 | 54.23 | 58.42 |
| Linear Probing (CVPR 22) | 79.79 | 70.02 | 46.99 | 46.48 | 71.16 | 52.28 | 57.39 |
| + Weight ens. (CVPR 22) | 79.80 | 70.45 | 48.41 | 47.89 | 73.00 | 53.07 | 58.56 (+1.17) |
| + VRF (NeurIPS 24) | 79.84 | 70.36 | 48.67 | 48.08 | 73.87 | 53.36 | 58.87 (+1.48) |
| + TIES (NeurIPS 23) | 79.75 | 70.33 | 48.25 | 48.32 | 73.78 | 53.13 | 58.76 (+1.37) |
| + DARE (ICML 24) | 79.14 | 70.26 | 48.14 | 47.74 | 73.11 | 53.01 | 58.45 (+1.06) |
| + MERGETUNE | 79.96 | 70.22 | 49.47 | 49.21 | 75.98 | 53.43 | 59.66 (+2.27) |
| + Weight ens. | 79.88 | 70.27 | 50.14 | 50.04 | 76.69 | 54.01 | **60.23** (+2.84) |
| E2E-FT (CVPR 22) | 81.31 | 70.61 | 45.12 | 36.62 | 65.63 | 50.51 | 53.70 |
| + Weight ens. (CVPR 22) | 82.51 | 73.11 | 51.62 | 47.61 | 75.13 | 55.71 | 60.64 (+6.94) |
| + VRF (NeurIPS 24) | 82.32 | 72.12 | 52.93 | 48.41 | 78.72 | 56.41 | 61.72 (+8.02) |
| + TIES (NeurIPS 23) | 82.27 | 72.84 | 51.67 | 47.81 | 74.48 | 54.87 | 60.33 (+6.63) |
| + DARE (ICML 24) | 81.09 | 70.21 | 45.79 | 35.55 | 65.23 | 50.13 | 53.38 (-0.32) |
| + MERGETUNE | 82.26 | 72.98 | 52.76 | 51.61 | 78.01 | 56.22 | 62.29 (+8.59) |
| + Weight ens. | 82.18 | 73.21 | 53.10 | 52.68 | 78.68 | 56.84 | **62.90** (+9.20) |

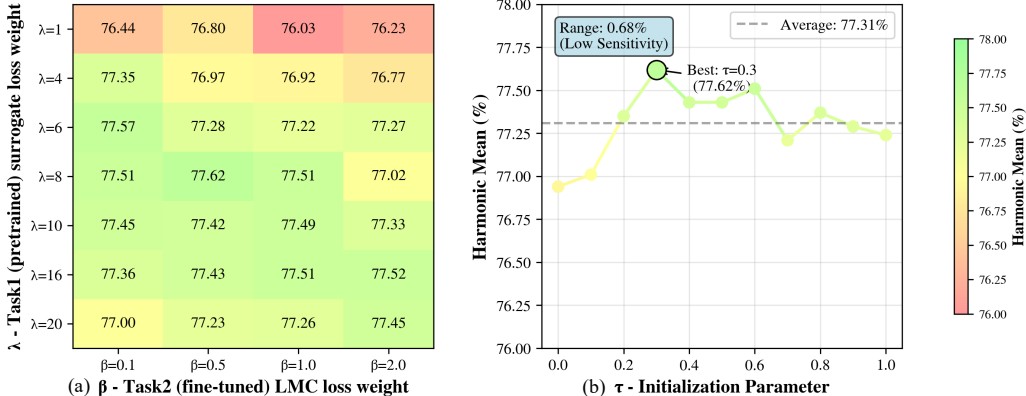

Figure 3: Hyperparameter sensitivity analysis: HM averaged over 11 datasets. (a) Surrogate loss weight $\lambda$ and Task2 LMC loss weight $\beta$. (b) Initialisation parameter $\tau$ of the continued model on performance (under $\lambda = 8.0$, $\beta = 0.5$), where $\tau = 0$ means using CLIP weights for initialisation and $\tau = 1$ means using the fine-tuned weights (e.g. KgCoOp here).

## 5.5 ID-OOD GENERALISATION

Table 4 presents the robust fine-tuning evaluation on ImageNet and five datasets under distribution shifts. We repurposed model merging methods for this evaluation. However, training-free merging methods, TIES and DARE, exhibit unstable performance and severe degradation under challenging shifts. Our MERGETUNE outperforms SOTA ensembling methods, such as VRF. Notably, MERGETUNE employs a single model for inference, eliminating the need for failure set construction or per-sample distance computations, which can incur additional inference costs as VRF did.

Furthermore, when combined with weight-space ensembling — which interpolates parameters between the zero-shot and our LMC-tuned checkpoint using a single mixing weight — our MERGETUNE achieves the highest average OOD accuracies: 60.23 (Linear Probing) and 62.90 (E2E-FT). This demonstrates that the LMC-tuned model is a superior replacement for standard fine-tuned checkpoints in ensemble schemes. These results establish our achievement on better ID-OOD trade-offs without complex design, while for the first time outperforming CLIP in all cases.

## 5.6 FURTHER ANALYSIS

We evaluate the sensitivity of MERGETUNE to key hyper-parameters, including surrogate loss weight $\lambda$, Task2 LMC loss weight $\beta$, and the initialisation of the continued model. We conduct this ablation averaged over 11 datasets using our method based on KgCoOp initialisation. Figure 3(a) demonstrates that our method is robust to hyperparameter choices, with harmonic mean performance varying within a narrow range across different $\lambda$ and $\beta$ combinations. Optimal perfor-

mance is achieved when $\lambda \in [8, 16]$ and $\beta \in [0.1, 0.5]$, resulting in balanced loss values between different terms. Figure 3(b) shows the impact of the initialisation of the parameters for continued fine-tuning. Given both fine-tuned and zero-shot checkpoints, we initialise the parameters by merging them as $w = (1 - \tau)\hat{w}_1 + \tau\hat{w}_2, \tau \in [0, 1]$, where $\tau = 0$ corresponds to CLIP weights and $\tau = 1$ to fine-tuned weights. When the initialisation combines CLIP and the fine-tuned model in a balanced manner ($\tau \in [0.3, 0.6]$), performance remains optimal.

# 6 CONCLUSION

We introduced continued fine-tuning (CFT), a new paradigm that recovers pretrained knowledge of a task-adapted VLM. Our method, MERGETUNE, searches for a continued model that is linear-mode connected to zero-shot and fine-tuned checkpoints, thus effectively integrating their knowledge. MERGETUNE is model-agnostic and can be applied post hoc to existing fine-tuned VLMs, without requiring architectural changes. Experimental results on base-to-novel, cross-dataset, domain generalisation, and robust fine-tuning benchmarks all showed the efficacy of our MERGETUNE, establishing continued fine-tuning as a promising direction for advancing VLM adaptation.

# 7 STATEMENT

## 7.1 ETHICS STATEMENT

This work focuses on methodological improvements for adapting vision-language models and does not involve the collection of new human subject data. All datasets used in our experiments are publicly available, widely adopted in prior research, and contain images and annotations curated for academic use. We follow the same usage protocols as the original dataset authors and cite the relevant sources. As our method builds on pretrained models such as CLIP, potential biases or societal impacts inherited from these models may still be present in downstream applications. We encourage careful consideration of these risks when deploying our approach in real-world systems.

## 7.2 REPRODUCIBILITY STATEMENT

We ensure reproducibility by providing comprehensive details of our experimental setup. The datasets, model architectures, and baselines we use are all publicly available. Our training configurations, including hyperparameters and optimization settings, are described in the main paper and supplement. To further support replication, we include our code in the supplementary material. In addition, we perform multiple runs with different random seeds and report averaged results to ensure statistical robustness.

# 8 ACKNOWLEDGMENTS

This research was supported by the UKRI-AHRC CoSTAR National Lab for Creative Industries Research and Development (AH/Y001060/1)

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

# A  APPENDIX

## A.1  DISCLOSURE OF LLM USAGE

Large language models were used to enhance grammar and address editorial issues. All ideas, experimental design, data analysis, and conclusions are the original work of the authors.

## A.2  EXPERIMENTAL SETTINGS

Our framework is implemented in PyTorch Paszke et al. (2019), and all experiments are conducted on an NVIDIA 3090 GPU.

**Few-shot setting**  To evaluate MERGETUNE, we apply it to the following base methods: the prompt learning methods CoOp (Zhou et al., 2022b) and KgCoOp (Yao et al., 2023), the adapter-based method MMA (Yang et al., 2024), and the distillation-based method PromptKD (Li et al., 2024). For each base method, we first reproduce the fine-tuning procedure to obtain a downstream checkpoint. Our MERGETUNE is then applied to this checkpoint in conjunction with the zero-shot CLIP model. Following the baselines, we adopt a standard data augmentation scheme of random resized cropping and flipping. We use Stochastic Gradient Descent (SGD) as the optimizer. The number of prompt context tokens is set to 4, initialised as *"a photo of a []"*. All few-shot experiments are trained with 16 shots.

**CoOp + MERGETUNE.** Following the baseline, we use a batch size of 128 and an initial learning rate of 0.002 for the first round of downstream training over 100 epochs. Our MERGETUNE is then trained for 50 epochs using the same optimiser and training schedule as the base method.

**KgCoOp + MERGETUNE.** KgCoOp follows the same training epochs, schedule, and data augmentation settings as CoOp. The regularisation weight is set to $\lambda=8.0$, as in the original configuration. Our MERGETUNE reuses the same training setup as the baseline.

**MMA + MERGETUNE.** Following the baseline, we insert the multi-modal unit from transformer block $k=5$ to the final block in both the language and vision branches, with a shared projection dimension of 32. In the base to novel generalisation, models are trained for 5 epochs with a batch size of 128 on ImageNet and 16 on the other ten datasets. For the other experimental settings, we set $k=9$ and train for only one epoch. Training is performed with mixed precision, and our MERGETUNE follows the same training schedule and number of epochs as the baseline for each dataset.

**PromptKD + MERGETUNE.** We use publicly available pre-trained teacher models and train student models for 20 epochs with a batch size of 8 and a learning rate of 0.005. Our MERGETUNE follows the same training schedule.

**Many-shot setting**  For robust finetuning, we apply MERGETUNE to both linear probing and end-to-end fine-tuning (E2E-FT). Same as in the few-shot setting, we first fine-tune CLIP to obtain finetuned checkpoints, and then apply MERGETUNE to derive results.

**Linear probing.** We train only the classifier head for 10 epochs using SGD with a learning rate of $1 \times 10^{-2}$, batch size $512$, and a cosine learning-rate schedule.

**End-to-end Fine-tuning.** We jointly update the encoder and classifier head for 10 epochs using AdamW with a learning rate of $3 \times 10^{-5}$, batch size $512$, weight decay 0.05, and a cosine learning rate schedule with warmup. After obtaining the fine-tuned checkpoint, we apply MERGETUNE for the same number of epochs, reusing the corresponding optimiser and training schedule.

**Datasets**  We evaluate the **few-shot setting** on various settings. For generalisation from *base-to-novel* classes and *cross-dataset evaluation*, we test our method on 11 diverse recognition datasets. Specifically, these include ImageNet-1K Deng et al. (2009) and Caltech-101 Fei-Fei et al. (2004) for generic object classification; OxfordPets Parkhi et al. (2012), StanfordCars Krause et al. (2013), Flowers-102 Nilsback & Zisserman (2008), Food-101 Bossard et al. (2014), and FGVCAircraft Maji et al. (2013) for fine-grained classification; SUN-397 Xiao et al. (2010) for scene recognition; UCF-101 Soomro et al. (2012) for action recognition; DTD Cimpoi et al. (2014) for texture classification;

and EuroSAT Helber et al. (2019) for satellite imagery recognition. For *domain generalisation* experiments, we use ImageNet-1K Deng et al. (2009) as the source dataset and its four variants as target datasets: ImageNet-V2 Recht et al. (2019), ImageNet-Sketch Wang et al. (2019), ImageNet-A Hendrycks et al. (2021b), and ImageNet-R Hendrycks et al. (2021a).

For the **many-shot setting**, we report results on ImageNet Deng et al. (2009) and its five distribution-shift variants: (1) ImageNet-V2 (-V2) Recht et al. (2019): test images sampled a decade after the original ImageNet. (2) ImageNet-R (-R) Hendrycks et al. (2021a): contains renditions (e.g., art, cartoons, graffiti). (3) ImageNet-Sketch (-S) Wang et al. (2019): consists of sketches instead of natural photos. (4) ImageNet-A (-A) Hendrycks et al. (2021b): real-world images that are misclassified by ResNet models. (5) ObjectNet Barbu et al. (2019): a test set featuring objects with diverse backgrounds, rotations, and viewpoints.

## A.3 PSEUDO-CODE

---
**Algorithm 1** Continued finetuning — MERGETUNE

---
1: **Input:** zero-shot weights $\hat{w}_1$, fine-tuned weights $\hat{w}_2$, downstream dataset $\mathcal{D}_2$, downstream task loss $\mathcal{L}_2(\cdot)$, surrogate loss weight $\lambda$, LMC loss weight $\beta$, number of interpolation points $n$, initialisation coefficient $\tau$
2: **Output:** continued model weights $w$
3: $A \leftarrow \{1/n, 2/n, \ldots, (n-1)/n\}$
4: Initialise $w \leftarrow (1-\tau)\hat{w}_1 + \tau\hat{w}_2$      ▷ $w$ is initialised by blending zero-shot and fine-tuned weights.
5: **for** each training epoch **do**
6:      **for** each minibatch $(x, y)$ from $\mathcal{D}_2$ **do**
7:          $\mathcal{L}_{\text{task}} \leftarrow \mathcal{L}_2(w; x, y)$      ▷ Downstream task loss
8:          $\mathcal{L}_{\text{sur}} \leftarrow \|w - \hat{w}_1\|^2$      ▷ Surrogate loss for Task 1
9:          $\mathcal{L}_{\text{LMC}} \leftarrow 0$
10:          **for** each $\alpha \in A$ **do**
11:              $w_{\text{interp}} \leftarrow \hat{w}_2 + \alpha(w - \hat{w}_2)$
12:              $\mathcal{L}_{\text{LMC}} \leftarrow \mathcal{L}_{\text{LMC}} + \mathcal{L}_2(w_{\text{interp}}; x, y)$
13:          **end for**
14:          $\mathcal{L}_{\text{LMC}} \leftarrow \mathcal{L}_{\text{LMC}}/|A|$      ▷ Average LMC loss
15:          $\mathcal{J} \leftarrow \mathcal{L}_{\text{task}} + \lambda\mathcal{L}_{\text{sur}} + \beta\mathcal{L}_{\text{LMC}}$
16:          Update $w$ with SGD step on $\mathcal{J}$
17:      **end for**
18: **end for**
19: **return** $w$

---

## A.4 ADDITIONAL EXPERIMENTS

### A.4.1 GENERALISATION TO OTHER BACKBONES

We also evaluate our MERGETUNE with other backbones to test its generalisation. We report results in Table 5 using CLIP ViT-B/32, which was commonly used in prior robust fine-tuning works (Zhu et al., 2024; Kim et al., 2024). The conclusion is consistent with the results in Table 4.

To further evaluate the scalability of MERGETUNE, we conduct experiments with three additional vision-language model backbones: CLIP-L/14, Siglip2-B/16, and Siglip2-L/16 (Tschannen et al., 2025). We compare the base CoOp method with CoOp+MERGETUNE across all 11 datasets for the base-to-novel generalisation evaluation. As shown in Table 6, our method consistently improves performance with new model scales and architectures, demonstrating strong scalability and model-agnostic effectiveness that generalises well beyond specific model architectures.

Table 5: ID-OOD generaisation accuracy of various methods on ImageNet and distribution shifts for CLIP ViT-B/32 in the robust fine-tuning evaluation. Avg-D = average over domain-shifted datasets.

| Method | Imagenet | Distribution shifts | | | | ObjectNet | Avg-D |
|---|---|---|---|---|---|---|---|
| | | -V2 | -S | -A | -R | | |
| Zero-shot (CLIP) | 63.34 | 55.94 | 42.33 | 31.45 | 69.26 | 43.46 | 48.49 |
| E2E-FT (CVPR 22) | 76.21 | 64.21 | 39.62 | 20.35 | 57.38 | 39.63 | 44.24 |
| + Weight ens. (CVPR 22) | 77.62 | 66.86 | 45.67 | 28.21 | 66.68 | 44.72 | 50.43 (+6.19) |
| + VRF (NeurIPS 24) | 77.60 | 66.70 | 47.00 | 29.20 | 70.90 | 46.30 | 52.02 (+7.78) |
| + TIES (NeurIPS 23) | 77.20 | 65.88 | 46.20 | 28.12 | 69.21 | 44.56 | 50.79 (+6.55) |
| + DARE (ICML 24) | 77.43 | 66.32 | 46.49 | 28.55 | 69.75 | 45.13 | 51.25 (+7.01) |
| + MERGETUNE | 77.81 | 67.02 | 46.83 | 31.80 | 70.67 | 47.01 | 52.67 (+8.43) |
| + Weight ens. | 77.27 | 66.59 | 46.87 | 32.40 | 71.09 | 46.99 | **52.79** (+8.55) |

Table 6: Base-to-novel generalisation experiments across different vision-language models on 11 datasets. Our method achieves consistent performance improvement over CoOp baseline across all model architectures.

| Method | Average | | | ImageNet | | | Caltech101 | | | OxfordPets | | |
|---|---|---|---|---|---|---|---|---|---|---|---|---|
| | Base | Novel | HM | Base | Novel | HM | Base | Novel | HM | Base | Novel | HM |
| *CLIP-L14* | | | | | | | | | | | | |
| CoOp | 86.61 | 74.69 | 80.21 | 82.01 | 71.80 | 76.57 | 98.66 | 96.07 | 97.35 | 96.08 | 98.36 | 97.21 |
| CoOp + MERGETUNE | 85.90 | 78.70 | 82.14 (+1.93) | 81.93 | 75.57 | 78.62 | 97.59 | 98.00 | 97.79 | 96.64 | 98.32 | 97.47 |
| *Siglip2-B16* | | | | | | | | | | | | |
| CoOp | 78.31 | 79.02 | 78.66 | 79.97 | 77.38 | 78.66 | 98.52 | 97.82 | 98.17 | 95.85 | 97.97 | 96.90 |
| CoOp + MERGETUNE | 80.75 | 79.79 | 80.26 (+1.60) | 80.03 | 77.18 | 78.58 | 98.56 | 97.82 | 98.19 | 96.12 | 98.04 | 97.07 |
| *Siglip2-L16* | | | | | | | | | | | | |
| CoOp | 81.56 | 83.31 | 82.43 | 84.01 | 80.33 | 82.13 | 98.34 | 98.03 | 98.19 | 97.79 | 98.71 | 98.25 |
| CoOp + MERGETUNE | 83.04 | 84.43 | 83.73 (+1.30) | 84.31 | 80.42 | 82.32 | 98.90 | 98.23 | 98.56 | 97.33 | 98.41 | 97.87 |

| Method | StanfordCars | | | Flowers102 | | | Food101 | | | FGVCAircraft | | |
|---|---|---|---|---|---|---|---|---|---|---|---|---|
| | Base | Novel | HM | Base | Novel | HM | Base | Novel | HM | Base | Novel | HM |
| *CLIP-L14* | | | | | | | | | | | | |
| CoOp | 82.72 | 82.89 | 82.81 | 98.96 | 75.04 | 85.35 | 93.46 | 93.55 | 93.51 | 52.64 | 32.95 | 40.53 |
| CoOp + MERGETUNE | 82.49 | 83.81 | 83.14 | 97.59 | 79.77 | 87.79 | 94.01 | 94.96 | 94.48 | 50.28 | 42.73 | 46.20 |
| *Siglip2-B16* | | | | | | | | | | | | |
| CoOp | 86.18 | 96.23 | 90.93 | 89.68 | 85.37 | 87.47 | 92.24 | 93.42 | 92.83 | 29.87 | 39.79 | 34.12 |
| CoOp + MERGETUNE | 86.24 | 96.26 | 90.98 | 92.40 | 86.33 | 89.27 | 92.49 | 93.51 | 93.00 | 31.39 | 40.19 | 35.25 |
| *Siglip2-L16* | | | | | | | | | | | | |
| CoOp | 90.34 | 97.58 | 93.82 | 92.72 | 87.28 | 89.91 | 94.75 | 95.51 | 95.13 | 35.91 | 54.63 | 43.34 |
| CoOp + MERGETUNE | 90.66 | 97.50 | 93.96 | 93.42 | 87.34 | 90.28 | 95.11 | 95.53 | 95.32 | 36.03 | 59.23 | 44.80 |

| Method | SUN397 | | | DTD | | | EuroSAT | | | UCF101 | | |
|---|---|---|---|---|---|---|---|---|---|---|---|---|
| | Base | Novel | HM | Base | Novel | HM | Base | Novel | HM | Base | Novel | HM |
| *CLIP-L14* | | | | | | | | | | | | |
| CoOp | 84.67 | 72.33 | 78.02 | 82.49 | 59.50 | 69.13 | 93.40 | 64.94 | 76.61 | 87.63 | 74.16 | 80.33 |
| CoOp + MERGETUNE | 85.87 | 73.31 | 79.10 | 80.82 | 62.45 | 70.46 | 92.33 | 77.20 | 84.09 | 85.37 | 79.55 | 82.41 |
| *Siglip2-B16* | | | | | | | | | | | | |
| CoOp | 83.27 | 70.81 | 76.54 | 77.35 | 69.81 | 73.39 | 56.75 | 58.17 | 57.45 | 71.72 | 82.40 | 76.69 |
| CoOp + MERGETUNE | 84.87 | 72.31 | 78.09 | 79.98 | 72.06 | 75.81 | 68.38 | 60.30 | 64.09 | 77.75 | 83.65 | 80.59 |
| *Siglip2-L16* | | | | | | | | | | | | |
| CoOp | 84.97 | 74.11 | 79.17 | 78.93 | 72.62 | 75.65 | 61.26 | 70.10 | 65.38 | 78.16 | 87.56 | 82.59 |
| CoOp + MERGETUNE | 85.87 | 75.16 | 80.16 | 81.10 | 78.26 | 79.65 | 70.56 | 70.59 | 70.58 | 80.17 | 88.05 | 83.92 |

## A.4.2 HYPERPARAMETER EVALUATION

We evaluate the sensitivity of MERGETUNE to key hyper-parameters, including surrogate loss weight $\lambda$, Task2 LMC loss weight $\beta$, and the initialisation parameter of the continued model. Here is the detailed experimental results. Detailed experimental results are provided in Table 7 and Table 8.

Table 7: Hyperparameter sensitivity analysis: all reported accuracies are averaged over 11 datasets, across surrogate loss weight $\lambda$ and Task2 LMC loss weight $\beta$.

| Parameters | | Performance (%) | | |
|---|---|---|---|---|
| $\lambda$ | $\beta$ | Avg Base Acc | Avg New Acc | Avg Harmonic Mean |
| 1 | 0.1 | 82.86 | 70.94 | 76.44 |
| 1 | 0.5 | 82.74 | 71.66 | 76.80 |
| 1 | 1 | 82.89 | 70.22 | 76.03 |
| 1 | 2 | 82.75 | 70.67 | 76.23 |
| 4 | 0.1 | 82.38 | 72.90 | 77.35 |
| 4 | 0.5 | 82.47 | 72.15 | 76.97 |
| 4 | 1 | 82.64 | 71.94 | 76.92 |
| 4 | 2 | 82.70 | 71.63 | 76.77 |
| 6 | 0.1 | 82.31 | 73.34 | 77.57 |
| 6 | 0.5 | 82.41 | 72.76 | 77.28 |
| 6 | 1 | 82.56 | 72.53 | 77.22 |
| 6 | 2 | 82.80 | 72.43 | 77.27 |
| 8 | 0.1 | 82.04 | 73.45 | 77.51 |
| 8 | 0.5 | 82.11 | 73.60 | **77.62** |
| 8 | 1 | 82.39 | 73.18 | 77.51 |
| 8 | 2 | 82.64 | 72.11 | 77.02 |
| 10 | 0.1 | 81.68 | 73.64 | 77.45 |
| 10 | 0.5 | 82.02 | 73.31 | 77.42 |
| 10 | 1 | 82.20 | 73.29 | 77.49 |
| 10 | 2 | 82.58 | 72.72 | 77.33 |
| 16 | 0.1 | 80.96 | 74.06 | 77.36 |
| 16 | 0.5 | 81.42 | 73.82 | 77.43 |
| 16 | 1 | 81.65 | 73.76 | 77.51 |
| 16 | 2 | 82.10 | 73.43 | 77.52 |
| 20 | 0.1 | 79.93 | 74.27 | 77.00 |
| 20 | 0.5 | 80.54 | 74.18 | 77.23 |
| 20 | 1 | 80.88 | 73.95 | 77.26 |
| 20 | 2 | 81.78 | 73.55 | 77.45 |

Table 8: Initialisation analysis of continued model: performance comparison across 11 datasets with $\lambda = 8.0$ and $\beta = 0.5$. Here, $\tau = 0$ corresponds to initialising the continued model with the CLIP weights, while $\tau = 1$ corresponds to the finetuned KgCoOp weights.

| $\tau$ | Base Acc | New Acc | Harmonic Mean |
|---|---|---|---|
| 0.0 | 81.24 | 73.07 | 76.94 |
| 0.1 | 81.47 | 73.01 | 77.01 |
| 0.2 | 81.84 | 73.33 | 77.35 |
| **0.3** | **82.11** | **73.60** | **77.62** |
| 0.4 | 81.91 | 73.41 | 77.43 |
| 0.5 | 82.01 | 73.34 | 77.43 |
| 0.6 | 82.10 | 73.42 | 77.51 |
| 0.7 | 82.07 | 72.90 | 77.21 |
| 0.8 | 82.15 | 73.13 | 77.37 |
| 0.9 | 82.03 | 73.07 | 77.29 |
| 1.0 | 82.16 | 72.87 | 77.24 |

### A.4.3 SENSITIVITY TO INTERPOLATION NUMBER $N_\alpha$

To better understand our MERGETUNE, we conducted additional analysis. In Table 9, we varied the number of interpolation points $N_\alpha$, which is used to approximate the LMC expectation term. Performance improves consistently as $N_\alpha$ increases from 1 to 10, after which it saturates. Meanwhile, the training time gradually increases with scaling $N_\alpha$ up: when using KgCoOp+MERGETUNE, the cost is about $3\times$ that of KgCoOp at $N_\alpha = 5$, increases to $5\times$ at $N_\alpha = 10$, and reaches $7\times$ at $N_\alpha = 15$. Thus, although $N_\alpha = 10$ provides a slight further improvement in accuracy, the associated overhead is more noticeable, making $N_\alpha = 5$ a more balanced choice.

Table 9: Effect of the number of interpolation points ($\alpha$) in the LMC approximation. Results are averaged over 11 datasets using MERGETUNE with KgCoOp weights. Performance improves steadily as $\alpha$ increases from 1 to 5, beyond which accuracy saturates and further increases in $\alpha$ only raise fine-tuning cost.

| Number of Interpolation Points ($N_\alpha$) | Base Acc | New Acc | Harmonic Mean |
|:---:|:---:|:---:|:---:|
| 1 | 81.75 | 73.12 | 77.19 |
| 3 | 82.01 | 73.51 | 77.53 |
| 5 | 82.11 | 73.60 | 77.62 |
| 10 | 82.13 | 73.71 | 77.69 |
| 15 | 82.13 | 73.72 | 77.70 |

### A.4.4 PERFORMANCE OF INTERPOLATED MODELS

We also evaluate the interpolated models between our continued model and the original zero-shot CLIP, as well as between it and the initially fine-tuned KgCoOp model on the base-to-novel setting. As shown in Figure 4, the results demonstrate smooth, low-loss (high-accuracy illustrated) paths between our continued model and both endpoints. Interpolation with the fine-tuned KgCoOp (blue line) shows stable, gradually improving base-class performance, while interpolation with zero-shot CLIP (orange line) maintains consistent, slightly improving novel-class performance throughout. The two smooth trajectories without performance fluctuations indicate that our continued model resides in a connected region of the loss landscape that spans both endpoints, confirming that MERGE-TUNE successfully establishes linear mode connectivity and effectively integrates knowledge from both the zero-shot and fine-tuned solutions.

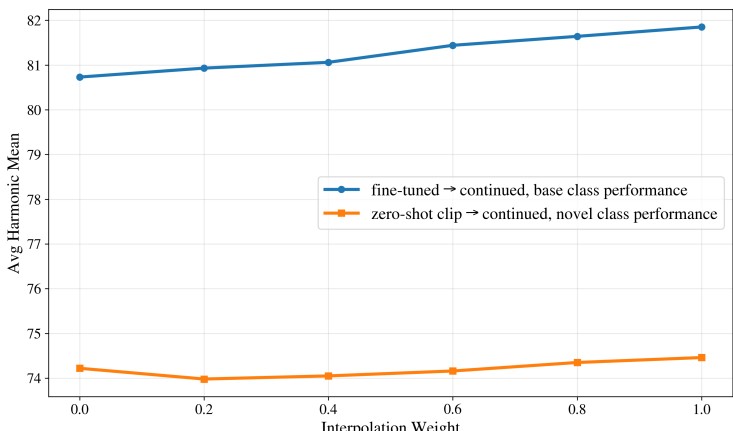

Figure 4: Linear mode connectivity analysis on base-to-novel generalisation. We interpolate between our continued model and both the zero-shot CLIP model (orange line, novel-class performance) and the fine-tuned KgCoOp model (blue line, base-class performance). The smooth paths confirm that MERGETUNE successfully establishes linear mode connectivity with both endpoints, maintaining strong performance throughout interpolation. Results are averaged over 11 datasets.

### A.4.5 WILL CONTINUED FINETUNING INTRODUCE OVER-MERGING?

To evaluate potential over-merging from our continued finetuning, we analysed MERGETUNE's performance across all 11 datasets, ranging from 10 to 100 epochs using CoOp + MERGETUNE

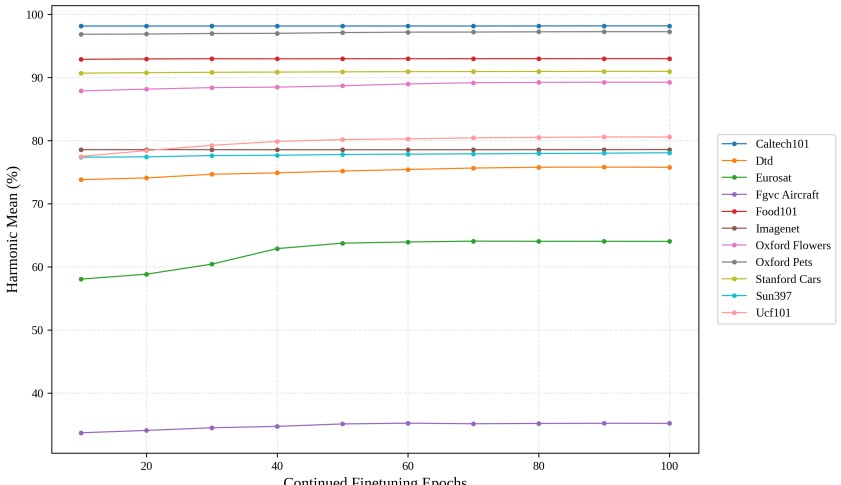

Figure 5: MERGETUNE exhibits no over-merging across extended training (ranging from 10 to 100 epochs). Performance trajectories across all 11 datasets show either stable performance or continuous improvement without degradation. No performance decline validates its robustness against over-merging through our dual-anchored linear mode connectivity objective.

on Siglip2-B/16 (as in Figure 5). Results exhibit two primary patterns: stable performance (on Caltech101, Food101, Oxford Pets, and ImageNet) and continuous improvement (on DTD, Oxford Flowers, Stanford Cars, UCF101, EuroSAT, and FGVC Aircraft). Across all datasets, no performance degradation is observed during continued finetuning, confirming that our linear-mode connectivity-guided model merging does not lead to over-merging, thanks to the effective regularisation applied to both zero-shot and fine-tuned endpoints.

