# OpenReview forum: "MergeTune: Continued Fine-Tuning of Vision-Language Models"
_ICLR.cc/2026/Conference — ICLR 2026 Poster_

### Official Review · Reviewer_Poyv · 2025-10-25

**Soundness:** 2
**Presentation:** 3
**Contribution:** 2
**Rating:** 4
**Confidence:** 4

**Summary:**

This paper addresses the problem of catastrophic forgetting in fine-tuned vision-language models (VLMs). The authors propose a new paradigm called Continued Fine-Tuning (CFT), which aims to recover pretrained knowledge after a model has already been adapted, rather than just mitigating forgetting during adaptation. Extensive experiments on CLIP are conducted to verify the effectiveness.

**Strengths:**

1. The paper is clearly written and easy to follow.

2. The experiments are comprehensive.

**Weaknesses:**

1. Some assumptions are overly strong and insufficiently discussed. The derivation of the second-order surrogate is not entirely convincing. For example, for smaller models such as ViT-B/16, it is questionable whether the gradients of the trained model are close to zero, as the authors assume. Additionally, the assumption
𝐻1≈μI appears too strong.

2. Comparing a training-free method with training-based methods seems unfair, as they operate under fundamentally different settings.

3. The performance improvement is limited. The proposed method requires significantly more training time to train a new model, yet the gains on downstream tasks are modest. In the base-to-new setting, it even underperforms compared to ATPrompt.

4. There is no discussion of training time or computational cost. It would be helpful to include an analysis of the training cost of the proposed method.

5. The paper claims that the proposed method is designed for vision-language models, yet only CLIP is evaluated. A broader experimental validation would strengthen this claim.

**Questions:**

See weaknesses.

---

> ### Author Response · Authors · 2025-11-19
>
> **W1**: Validate the surrogate loss assumptions
>
> **R**: Thank you for raising this point. We address both assumptions separately.
>
> **Assumption 1: Gradient near zero ($\nabla \mathcal{L}_1(\hat{w}_1) \approx 0$).**
> This assumption is reasonable because assuming pretrained checkpoints reside near local optima is standard practice. However, since CLIP was pretrained on an unpublished dataset, we cannot access to verify this directly. To provide empirical support, we evaluated the gradient norm using ImageNet-1k validation set as a proxy, which contains diverse classes. For CLIP ViT-B/16, we obtained a **relative gradient norm** of **0.078**, indicating the model is near a stationary point. More importantly, our method's **consistent improvements across all baselines** empirically demonstrate that this approximation does not adversely affect MergeTune's effectiveness.
>
> **Assumption 2: Isotropic Hessian ($H_1 \approx \mu I$).** We empirically validated this assumption by comparing it with the **Fisher Information Matrix (FIM)**, which provides an anisotropic approximation to the Hessian via $\mathcal{R}_{\text{Task1}} = (w - \hat{w}_1)^T F (w - \hat{w}_1)$, where $F = \mathbb{E}[\nabla \log p(y|x; w) \nabla \log p(y|x; w)^T]$. This comparison directly tests sensitivity to violations of the isotropic assumption. **Results show MergeTune is not sensitive to this assumption**: on base-to-novel generalization (11 datasets average), KgCoOp + MergeTune with the isotropic assumption ($H_1 \approx \mu I$) achieves **77.98% HM**, while the theoretically more accurate FIM achieves only **77.75% HM**.
>
> This proves the isotropic assumption $H_1 \approx \mu I$ is well-suited for our linear mode connectivity objective, providing uniform regularization that maintains proximity to the zero-shot model while allowing the LMC term full flexibility to discover smooth, low-loss paths connecting both endpoints. Moreover, this approach is computationally efficient, requiring only a scalar hyperparameter $\lambda$ rather than computing and storing full curvature information. The consistent improvements achieved by MergeTune across all experiments (Tables 1-4) empirically validate that this isotropic assumption works effectively in practice for LMC-guided merging.
>
> **W2**: Unfair comparison with the training-free methods
>
> **R**: Training-free model merging methods (TIES, DARE) is **one category of our compared methods**. The comparison demonstrates that simply merging weights without additional guidance is insufficient for recovering forgotten knowledge, validating the necessity of our continued fine-tuning approach. However, we compared our method with other more sophisticated merging methods, such as Wise-FT and VRF, and demonstrated the promising performance of MergeTune. We also welcome additional relevant baselines if the reviewer has specific suggestions.
>
> **W3**: Modest performance gains, and MergeTune underperforms ATPrompt?
>
> **R**: We thank the reviewer for this question. We respectfully disagree that the gains are modest – our improvements are substantial and consistent across all settings.
>
> **Base-to-novel generalisation.** MergeTune outperforms CoOp by 5.58% HM, KgCoOp by 0.97%, MMA by 0.57%, and PromptKD by 0.36%. The gain magnitude correlates with the degree of forgetting in each method – those suffering more forgetting benefit more – showing that MergeTune reliably recovers lost pretrained knowledge.
> **Cross-dataset generalisation & robust fine-tuning.** Across all baselines, MergeTune yields consistent gains (+2.27% to +8.59% Avg-D).
>
> Regarding the comparison to ATPrompt: it is worthy noted that ATPrompt (ICCV 2025) uses external **LLM models** for attribute mining to reach +2.99% on CoOp, while our MergeTune achieves +5.58% purely from the base + fine-tuned checkpoints – no external knowledge required. Moreover, our MergeTune is a post-hoc method applicable to most VLM fine-tuning methods and can be applied to ATPrompt-enhanced models for potentially further improvements.

---

> > ### Author Response · Authors · 2025-11-19
> >
> > **W4**: Training time or computational cost
> >
> > **R**: Thank you for raising the concern. The additional computational overhead of MergeTune mainly comes from the continued finetuning. Additionally, the training overhead scales with the number of interpolation points $N_\alpha$ used in the LMC regularizer (as analysed in **Table 9** of the updated PDF). Generally, the overhead is approximately $3\times$ that of the base fine-tuning method when $N_\alpha=5$, increases to $5\times$ when $N_\alpha=10$, and reaches $7\times$ when $N_\alpha=15$.
> >
> > While this looks substantial relative to base VLM fine-tuning alone, it is important to contextualise this overhead. Pretraining vision-language models like CLIP requires enormous computational resources – CLIP was trained on 400 million image-text pairs using 256 V100 GPUs for approximately two weeks, equivalent to thousands of GPU-hours. In contrast, fine-tuning time is considerably shorter – for example, fine-tuning CLIP on ImageNet with a single A100 GPU typically requires less than one hour. Therefore, even a $3\times$ increase ($N_\alpha=5$) translates to only 2-3 more hours of training.
> >
> > This modest additional cost is justified because MergeTune's objective is to **restore the pretrained knowledge** acquired during the expensive pretraining phase. By recovering forgotten capabilities through continued fine-tuning, we ensure that the substantial computational investment in pretraining is not wasted due to catastrophic forgetting. The overhead is thus a worthwhile investment to maintain the model's generalisation abilities while adapting to downstream tasks.
> >
> > **W5**: Generalisation ability to larger backbones or non-CLIP VLMs
> >
> > **R**: Thank you for this suggestion. To demonstrate MergeTune's scalability, we have now conducted additional experiments on larger vision-language models CLIP ViT-L/14, and with a different VLM family SigLIP [1]. **Table 6** in the updated PDF presents comprehensive base-to-novel generalisation results across three model architectures for all 11 datasets. The following summary shows averaged performance over 11 datasets:
> >
> > - **CLIP-L/14** (CLIP large): CoOp + MergeTune achieves 82.14% HM (+1.93% over CoOp baseline 80.21%)
> > - **SigLIP2-B/16** (a different VLM backbone): CoOp + MergeTune achieves 80.26% HM (+1.60% over CoOp baseline 78.66%)
> > - **SigLIP2-L/16** (SigLIP large): CoOp + MergeTune achieves 83.73% HM (+1.30% over CoOp baseline 82.43%)
> >
> > These results demonstrate that **MergeTune consistently improves performance across different model scales and architectures**, validating its generalizability beyond CLIP ViT-B/16 and CLIP.
> >
> > [1] Tschannen M, Gritsenko A, Wang X, et al. Siglip 2: Multilingual vision-language encoders with improved semantic understanding, localization, and dense features. arXiv:2502.14786, 2025.

---

> > > ### Comment · Reviewer_Poyv · 2025-11-20
> > > **Thank you for the rebuttal**
> > >
> > > I thank the authors for the detailed rebuttal. I have some further questions for the rebuttal:
> > >
> > > (1). "assuming pretrained checkpoints reside near local optima is standard practice". Is there any paper support this claim? I do not think that 0.078 is a small number near 0 (Is the gradient norm is L2 norm?).
> > >
> > > (2). For performance gains, although MergeTune outperforms KgCoOp by 0.97%, MMA by 0.57%, and PromptKD by 0.36% (not very significant to me), it also requires significantly more training time—several times longer.
> > >
> > > Other concerns are well addressed.

---

> > > > ### Author Response · Authors · 2025-11-21
> > > > **Response to Reviewer Poyv**
> > > >
> > > > Thank you for your follow-up questions.
> > > >
> > > > **Q1**: "assuming pretrained checkpoints reside near local optima is standard practice". Is there any paper support this claim? I do not think that 0.078 is a small number near 0 (Is the gradient norm is L2 norm?).
> > > >
> > > > **R:**  Literatures for supporting the assumption: This type of assumption has been used in the literature. For instance, BRECQ (ICLR 2021) [1] states: "Given the pre-trained model is converged to a minimum, the gradients can be safely thought to be close to 0," and SQuant (ICLR 2022) [2] similarly writes: "since a well-trained model has already converged, the gradient term will be close to 0 and thus can be safely ignored," omitting the first-order term in their Taylor expansions.
> > > >
> > > > Regarding the 0.078 gradient norm: Yes, the gradient norm is the L2 norm. We clarify that our assumption $\nabla \mathcal{L}_1(\hat{w}_1) \approx 0$ applies to the pretrained CLIP on the pretraining dataset. Since CLIP's pretraining dataset is not publicly accessible, we use ImageNet-1k as a proxy to measure the minima. This serves only as a reference, as gradient measures could differ across dataset proxies. As CLIP was not directly pretrained on ImageNet, this value can be considered in a reasonable range.
> > > >
> > > > Notably, our consistent gains across all base methods on various settings (Tables 1-4) demonstrate that MergeTune effectively recovers pretrained knowledge despite this approximation. All these suggest our approximation is appropriate for practical effectiveness.
> > > >
> > > > **References:**
> > > >
> > > > [1] Li Y, Gong R, Tan X, et al. BRECQ: Pushing the limit of post-training quantization by block reconstruction. ICLR 2021.
> > > >
> > > > [2] Guo C, Qiu Y, Leng J, et al. SQuant: On-the-fly data-free quantization via diagonal hessian approximation. ICLR 2022.
> > > >
> > > >
> > > >
> > > > **Q2**: For performance gains, although MergeTune outperforms KgCoOp by 0.97%, MMA by 0.57%, and PromptKD by 0.36% (not very significant to me), it also requires significantly more training time - several times longer.
> > > >
> > > > **R:** We note that, these base methods (KgCoOp, MMA, PromptKD) already employed different mechanisms – knowledge regularization, multi-modal adapters, and teacher model distillation – to mitigate their catastrophic forgetting. Achieving further improvements over these well-equipped methods is non-trivial. Also, the improvements correlate with severity of forgetting, which directly aligns with our motivation of restoring pretrained knowledge.
> > > >
> > > > As noted in the response to **Q2 of Reviewer wRZr**, even with 3-5x overhead, the continued finetuning on ImageNet only takes 2-3 hours on a single GPU. This extra cost is negligible compared to the massive computational cost of the VLM pretraining, but restores the VLM’s pretrained knowledge consistently and effectively.
> > > >
> > > > Furthermore, unlike ensemble methods (VRF, Wise-FT), MergeTune produces a single model, resulting in no test-time inference overhead, which effectively compensates for the additional training cost.

---

> > > > > ### Comment · Reviewer_Poyv · 2025-11-25
> > > > >
> > > > > Thank you for further clarification. My concerns are addressed. I will raise my score to 6.

---

> > > > > > ### Author Response · Authors · 2025-11-25
> > > > > > **Response to Reviewer Poyv**
> > > > > >
> > > > > > Thank you for raising your score. We sincerely appreciate your reviews – they have been very helpful in strengthening our work.

---

### Official Review · Reviewer_uAR8 · 2025-10-28

**Soundness:** 2
**Presentation:** 3
**Contribution:** 3
**Rating:** 6
**Confidence:** 5

**Summary:**

The paper proposes MERGETUNE, a novel *continued fine-tuning* (CFT) framework that recovers pretrained knowledge in vision-language models (VLMs) after standard fine-tuning has already occurred. Unlike prior work that attempts to mitigate forgetting during adaptation, MERGETUNE operates post hoc by leveraging linear mode connectivity (LMC) to find a model that maintains low-loss interpolation paths to both the original zero-shot model (e.g., CLIP) and the fine-tuned model (e.g., CoOp). To avoid the infeasible requirement of replaying the massive pretraining data, the authors introduce a second-order surrogate loss that approximates the LMC constraint for the zero-shot model using a simple ℓ² regularizer toward the pretrained weights. The method is model-agnostic, requires no architectural changes, and consistently improves performance across multiple VLM adaptation strategies (prompt-based, adapter-based, linear probing, full fine-tuning) and evaluation protocols (base-to-novel, cross-dataset, domain generalization, robust fine-tuning). Notably, MERGETUNE achieves a +5.6% harmonic mean gain over CoOp without adding parameters and outperforms CLIP in all evaluated settings, with further gains possible via ensembling.

**Strengths:**

(1) Introduces a conceptually fresh and practical paradigm—continued fine-tuning—that decouples adaptation from knowledge recovery, enabling post hoc enhancement of any existing fine-tuned VLM.
(2) Proposes a theoretically grounded yet simple method (MERGETUNE) based on linear mode connectivity, with a clever second-order surrogate that eliminates the need for pretraining data replay—a major practical bottleneck.
(3) Demonstrates consistent and significant improvements across diverse adaptation methods (CoOp, MMA, PromptKD, etc.) and evaluation settings (few-shot, many-shot, OOD robustness), validating the generality of the approach.
(4) Provides comprehensive empirical analysis, including ablation studies on hyperparameters, comparisons with training-free merging (TIES, DARE) and ensembling baselines (VRF, Wise-FT), and shows MERGETUNE reduces inference cost while improving performance.

**Weaknesses:**

(1) The surrogate loss assumes the Hessian of the pretraining loss at the zero-shot checkpoint is isotropic (H₁ ≈ μI) and that the gradient is near zero. While common, this may not hold for large-scale VLMs like CLIP trained on noisy web data. Could the authors provide empirical validation of these assumptions (e.g., via Hessian spectrum estimation on a subset) or discuss how violations might affect MERGETUNE’s performance?
(2) In Table 1, MERGETUNE improves CoOp by +5.58% HM but only +0.36% on PromptKD. The paper attributes this to PromptKD already preserving more knowledge. However, is it possible that MERGETUNE’s ℓ² regularizer toward CLIP inadvertently conflicts with PromptKD’s distillation objective, limiting gains? An analysis of the weight trajectory or feature similarity between CLIP, PromptKD, and MERGETUNE-enhanced PromptKD would help clarify this.
(3) The method samples a small number of α values (e.g., 5–10) to approximate the expectation in the LMC loss. How sensitive are results to the number and spacing of these α points? Did the authors experiment with adaptive sampling or importance weighting along the interpolation path to better capture high-loss regions?
(4) For adapter-based methods like MMA, which modify model architecture, MERGETUNE is applied only to the linear head (per Section 5.2). This seems inconsistent with the claim of being “model-agnostic.” Can MERGETUNE be applied to the full adapter parameters? If not, what architectural constraints limit its applicability, and how does partial application affect knowledge recovery?
(5) The continued model is initialized as a convex combination of w̃₁ and w̃₂ (τ ∈ [0,1]). Figure 3(b) shows optimal τ ∈ [0.3, 0.6], but this requires tuning. In real-world deployment, the ideal τ may be unknown. Does MERGETUNE include a practical strategy for selecting τ without validation data (e.g., based on fine-tuning loss magnitude or forgetting estimates)?
(6) All experiments use CLIP ViT-B/16 (and briefly ViT-B/32). How does MERGETUNE scale to larger backbones (e.g., ViT-L/14) or non-CLIP VLMs (e.g., ALIGN, BLIP)? Given that mode connectivity properties can vary with model size and training data, it would strengthen the paper to show results beyond the standard CLIP setting.

**Questions:**

Please weakness.

---

> ### Author Response · Authors · 2025-11-19
>
> **W1**: Validate the surrogate loss assumptions.
>
> **R**: Thank you for raising this point. We address both assumptions separately.
>
> **Assumption 1: Gradient near zero ($\nabla \mathcal{L}_1(\hat{w}_1) \approx 0$).**
> This assumption is reasonable because assuming pretrained checkpoints reside near local optima is standard practice. However, since CLIP was pretrained on an unpublished dataset, we cannot access to verify this directly. To provide empirical support, we evaluated the gradient norm using ImageNet-1k validation set as a proxy, which contains diverse classes. For CLIP ViT-B/16, we obtained a **relative gradient norm** of **0.078**, indicating the model is near a stationary point. More importantly, our method's **consistent improvements across all baselines** empirically demonstrate that this approximation does not adversely affect MergeTune's effectiveness.
>
> **Assumption 2: Isotropic Hessian ($H_1 \approx \mu I$).** We empirically validated this assumption by comparing it with the **Fisher Information Matrix (FIM)**, which provides an anisotropic approximation to the Hessian via $\mathcal{R}_{\text{Task1}} = (w - \hat{w}_1)^T F (w - \hat{w}_1)$, where $F = \mathbb{E}[\nabla \log p(y|x; w) \nabla \log p(y|x; w)^T]$. This comparison directly tests sensitivity to violations of the isotropic assumption. **Results show MergeTune is not sensitive to this assumption**: on base-to-novel generalization (11 datasets average), KgCoOp + MergeTune with the isotropic assumption ($H_1 \approx \mu I$) achieves **77.98% HM**, while the theoretically more accurate FIM achieves only **77.75% HM**.
>
> This proves the isotropic assumption $H_1 \approx \mu I$ is well-suited for our linear mode connectivity objective, providing uniform regularization that maintains proximity to the zero-shot model while allowing the LMC term full flexibility to discover smooth, low-loss paths connecting both endpoints. Moreover, this approach is computationally efficient, requiring only a scalar hyperparameter $\lambda$ rather than computing and storing full curvature information. The consistent improvements achieved by MergeTune across all experiments (Tables 1-4) empirically validate that this isotropic assumption works effectively in practice for LMC-guided merging.
>
>
> **W2**: Are PromptKD and MergeTune in conflict?
>
> **R**: Thank you for this insightful question. We computed Centred Kernel Alignment (CKA) feature similarity between CLIP, PromptKD, and PromptKD+MergeTune across various datasets.
>
> | Comparison | Average Similarity |
> |------------|------------------|
> | CLIP vs PromptKD | 0.6849 |
> | CLIP vs PromptKD+MergeTune | 0.6963 |
> | PromptKD vs PromptKD+MergeTune | 0.9958 |
>
> The CKA similarity between PromptKD and PromptKD+MergeTune consistently exceeds **0.99**, indicating near-perfect preservation of learned features.
> MergeTune exhibits a slight increase in similarity to CLIP, demonstrating controlled knowledge recovery without disrupting PromptKD's representations. This analysis **show no conflict** between MergeTune's regularizer and PromptKD's distillation objective. The modest improvement (+0.36% HM in Table 1) attributed to the fact that PromptKD already preserves pre-trained knowledge effectively through distillation, leaving minimal forgotten knowledge for MergeTune to recover.
>
>
> **W3**: How sensitive are results to the number and spacing of α sampling points? Ever tried adaptive sampling to focus on high-loss regions?
>
> **R**: Thank you for this question.
>
> We ablated the number of α points in **Table 9** in the updated PDF. Performance saturates beyond $N _ \alpha=5$: harmonic mean increases from 77.19% at $N _ \alpha=1$ to 77.62% at $N _ \alpha=5$, with only marginal gains to 77.69% at $N _ \alpha=10$ and 77.70% at $N _ \alpha=15$.
>
> We use uniform spacing, which has proven sufficient – saturation suggests that the loss landscape is well-captured. We didn't implement adaptive sampling because: (1) uniform sampling already achieves strong performance across all baselines, (2) our method explicitly seeks linear low-loss paths where uniform sampling is theoretically motivated, and (3) adaptive sampling would add computational overhead. We acknowledge that adaptive sampling strategies could potentially offer further improvements, particularly for non-uniform loss landscapes. Exploring importance-weighted or adaptive α selection represents an interesting direction for future work.

---

> > ### Author Response · Authors · 2025-11-19
> >
> > **W4**: Can MMA only be applied to merge linear heads?
> >
> > **R**: Thank you for this question. We apologise for the confusion in Section 5.2. The description of merging only the linear heads specifically refers to the training-free baseline methods (TIES and DARE), not to our MergeTune. Training-free merging methods require identical model structures to directly interpolate weights, which is why we could only merge the linear heads for MMA while keeping its adapter-modified image encoder fixed.
> >
> > In contrast, our MergeTune does not have this structural constraint – it is applied to MMA's full set of trainable adapter modules. This flexibility to handle arbitrary trainable parameters, regardless of architectural modifications, is precisely what makes our method model-agnostic.
> >
> > **W5**: Should optimal τ be selected for real-world deployment?
> >
> > **R**: Thank you for this valuable suggestion. In our experiments, we found that the range τ ∈ [0.3, 0.6] consistently and stably performs across all 11 datasets (Figure 3b), with minimal variation in the harmonic mean (< 0.4%). **This stability across 11 diverse benchmarks, which is already very robust,** suggests that τ = 0.5 serves as a reliable default initialisation for practitioners without requiring dataset-specific tuning or validation data. We recommend this simple default in practice.
> >
> > We acknowledge that developing more sophisticated automatic selection strategies based on fine-tuning loss magnitude or forgetting estimates, as you suggested, represents an interesting and valuable direction for future work.
> >
> > **W6**: Generalisation ability to larger backbones or non-CLIP VLMs
> >
> > **R**: Thank you for this suggestion. To demonstrate MergeTune's scalability, we have now conducted additional experiments on larger vision-language models CLIP ViT-L/14, and with a different VLM family SigLIP [1]. **Table 6** in the updated PDF presents comprehensive base-to-novel generalisation results across three model architectures for all 11 datasets. The following summary shows averaged performance over 11 datasets:
> >
> > - **CLIP-L/14** (CLIP large): CoOp + MergeTune achieves 82.14% HM (+1.93% over CoOp baseline 80.21%)
> > - **SigLIP2-B/16** (a different VLM backbone): CoOp + MergeTune achieves 80.26% HM (+1.60% over CoOp baseline 78.66%)
> > - **SigLIP2-L/16** (SigLIP large): CoOp + MergeTune achieves 83.73% HM (+1.30% over CoOp baseline 82.43%)
> >
> > These results demonstrate that **MergeTune consistently improves performance across different model scales and architectures**, validating its generalizability beyond CLIP ViT-B/16 and CLIP.
> >
> > [1] Tschannen M, Gritsenko A, Wang X, et al. Siglip 2: Multilingual vision-language encoders with improved semantic understanding, localization, and dense features. arXiv:2502.14786, 2025.

---

### Official Review · Reviewer_wRZr · 2025-10-30

**Soundness:** 3
**Presentation:** 3
**Contribution:** 2
**Rating:** 6
**Confidence:** 3

**Summary:**

This paper proposes MERGETUNE, a continued fine-tuning framework for vision-language models such as CLIP. Instead of preventing catastrophic forgetting during adaptation, the authors attempt to recover pre-trained knowledge after fine-tuning has already taken place. The key idea is to exploit linear mode connectivity (LMC) to find new parameters with two low-loss connections to the zero-shot and fine-tuned checkpoints, effectively merging their knowledge. A second-order surrogate term approximates the pretraining loss to avoid data replay from the pretraining task. The method is model-agnostic and post-hoc, requiring no architectural modifications. Experiments on base-to-novel, cross-dataset, domain generalization, and robust fine-tuning benchmarks show consistent improvements across CoOp, KgCoOp, MMA, and PromptKD, achieving state-of-the-art results with minimal overhead. Nevertheless, I am not an expert in VLMs, so the following comments and questions could be based on potentially inaccurate understandings or biased perspectives.

**Strengths:**

Strengths:

1. Introduces a novel post-hoc fine-tuning paradigm focusing on knowledge restoration instead of prevention, offering a new conceptual direction.

2. The approach is simple, elegant, and general, requiring no model modifications and applying to various adaptation methods.

3. Using linear mode connectivity as an explicit optimization objective provides solid geometric intuition and interpretability.

4. Experimental evaluation is extensive and convincing, demonstrating consistent gains across diverse datasets and settings.

**Weaknesses:**

1. The theoretical justification is limited, and it is unclear why continued fine-tuning converges to a mode-connected region.

2. The final objective closely resembles standard $L_2$-regularized fine-tuning, so the novelty might be overstated.

3. The isotropic Hessian assumption in the surrogate loss is strong and unvalidated; its practical effect remains unclear.

4. The paper does not quantify the extra training cost or test scalability on larger or multimodal models.

**Questions:**

1. How sensitive is MERGETUNE to violations of the isotropic curvature assumption ($H_1 \approx \mu I$)?

2. What is the additional training or computational overhead compared to standard fine-tuning?

3. Could continued fine-tuning lead to “over-merging,” where downstream adaptation performance is harmed after too many epochs?

---

> ### Author Response · Authors · 2025-11-19
>
> **W1**: The theoretical justification is limited, and it is unclear why continued fine-tuning converges to a mode-connected region.
>
> **R**: Thanks for raising this question. In the prior work, Entezari et al. [1] prove that different minima in overparameterized neural networks are inherently connected by low-loss linear paths -- i.e. linear mode-connected paths. The theoretical foundation is evidenced by Mirzadeh et al. [2], who demonstrate that explicitly training with linear mode connectivity as an objective yields solutions that preserve performance across multiple tasks. Recent work by Huang et al. [3] demonstrates that effective model merging requires optimisation to find the proper alignment between models, rather than heuristic averaging, as training-free methods (TIES, DARE) do. All the above demonstrate that mode connectivity is a known phenomenon and can be explicitly leveraged during model training. Empirical analysis in **Figure 4** in the updated supplementary also validates this conclusion.
>
> [1] Entezari et al., "The Role of Permutation Invariance in Linear Mode Connectivity of Neural Networks", ICLR 2022
>
> [2] Mirzadeh et al., "Linear Mode Connectivity in Multitask and Continual Learning", ICLR 2021
>
> [3] Ainsworth et al., "Git Re-Basin: Merging Models modulo Permutation Symmetries", ICLR 2023
>
>
> **W2**: Resembles standard $L_2$-regularized fine-tuning?
>
> **R**: Thank you for raising this point. While our final objective contains an $L_2$ regularisation term, it is not equivalent to standard $L_2$-regularised fine-tuning. Our method introduces a distinct linear-mode-connectivity (LMC) regularizer, which has not been previously introduced in conventional VLM fine-tuning methods. Overall, this final objective introduces the following differences: 1) it is a surrogate derived from our learning-based model-merging formulation; 2) it promotes linear-mode connectivity in weight space to two different model solutions, zero-shot and finetuned, rather than simply shrinking parameters towards the initial zero-shot model; 3) it leads to empirically different behaviour, as evidenced by our comparisons with $L_2$-regulariser-based methods, such as KgCoOp, in the experiments section.
>
>
> **W3** & **Q1**: The isotropic Hessian assumption.
>
> **R**: Thank you for raising this concern. We empirically validated this assumption by comparing it with the **Fisher Information Matrix (FIM)**, which provides an anisotropic approximation to the Hessian via $\mathcal{R}_{\text{Task1}} = (w - \hat{w}_1)^T F (w - \hat{w}_1)$, where $F = \mathbb{E}[\nabla \log p(y|x; w) \nabla \log p(y|x; w)^T]$. This comparison directly tests sensitivity to violations of the isotropic assumption. **Results show MergeTune is not sensitive to this assumption**: on base-to-novel generalization (11 datasets average), KgCoOp + MergeTune with the isotropic assumption ($H_1 \approx \mu I$) achieves **77.98% HM**, while the theoretically more accurate FIM achieves only **77.75% HM**.
>
> This proves the isotropic assumption $H_1 \approx \mu I$ is well-suited for our linear mode connectivity objective, providing uniform regularization that maintains proximity to the zero-shot model while allowing the LMC term full flexibility to discover smooth, low-loss paths connecting both endpoints. Moreover, this approach is **computationally efficient**, requiring only a scalar hyperparameter $\lambda$ rather than computing and storing full curvature information. The consistent improvements achieved by MergeTune across all experiments (Tables 1-4) empirically validate that this isotropic assumption works effectively in practice for LMC-guided merging.

---

> ### Author Response · Authors · 2025-11-19
>
> **W4**: The paper does not quantify the extra training cost or test scalability on larger or multimodal models.
>
> **R**: Thank you for raising this point about training cost and scalability.
>
> **Extra training cost:** Please refer to the response to **Q2**.
>
> **Scalability:**  To demonstrate MergeTune's scalability, we have now conducted additional experiments on larger vision-language models CLIP ViT-L/14, and with a different VLM family SigLIP [1]. **Table 6** in the updated PDF presents comprehensive base-to-novel generalisation results across three model architectures for all 11 datasets. The following summary shows averaged performance over 11 datasets:
>
> - **CLIP-L/14** (CLIP large): CoOp + MergeTune achieves 82.14% HM (+1.93% over CoOp baseline 80.21%)
> - **SigLIP2-B/16** (a different VLM backbone): CoOp + MergeTune achieves 80.26% HM (+1.60% over CoOp baseline 78.66%)
> - **SigLIP2-L/16** (SigLIP large): CoOp + MergeTune achieves 83.73% HM (+1.30% over CoOp baseline 82.43%)
>
> These results demonstrate that **MergeTune consistently improves performance across different model scales and architectures**, validating its generalizability beyond CLIP ViT-B/16 and CLIP.
>
> [1] Tschannen M, Gritsenko A, Wang X, et al. Siglip 2: Multilingual vision-language encoders with improved semantic understanding, localization, and dense features. arXiv:2502.14786, 2025.
>
>
> **Q2**: Computational overhead compared to standard fine-tuning?
>
> **R**: Thank you for raising the concern. The additional computational overhead of MergeTune mainly comes from the continued finetuning. Additionally, the training overhead scales with the number of interpolation points $N_\alpha$ used in the LMC regularizer (as analysed in **Table 9** of the updated PDF). Generally, the overhead is approximately $3\times$ that of the base fine-tuning method when $N_\alpha=5$, increases to $5\times$ when $N_\alpha=10$, and reaches $7\times$ when $N_\alpha=15$.
>
> While this looks substantial relative to base VLM fine-tuning alone, it is important to contextualise this overhead. Pretraining vision-language models like CLIP requires enormous computational resources – CLIP was trained on 400 million image-text pairs using 256 V100 GPUs for approximately two weeks, equivalent to thousands of GPU-hours. In contrast, fine-tuning time is considerably shorter – for example, fine-tuning CLIP on ImageNet with a single A100 GPU typically requires less than one hour. Therefore, even a $3\times$ increase ($N_\alpha=5$) translates to only 2-3 more hours of training.
>
> This modest additional cost is justified because MergeTune's objective is to **restore the pretrained knowledge** acquired during the expensive pretraining phase. By recovering forgotten capabilities through continued fine-tuning, we ensure that the substantial computational investment in pretraining is not wasted due to catastrophic forgetting. The overhead is thus a worthwhile investment to maintain the model's generalisation abilities while adapting to downstream tasks.
>
>
> **Q3**: Will many epochs of continued fine-tuning lead to “over-merging”?
>
> **R**: Thanks for raising this concern.
>
> We conducted a further analysis of MergeTune's behaviour across different training durations (10 to 100 epochs) on various datasets. **Figure 5** in the updated supplementary visualises these results. Across all datasets and up to 100 epochs, performance either improves or remains stable, demonstrating no over-merging.
>
> The linear mode connectivity (LMC) objective inherently prevents over-merging through dual constraints: (1) the surrogate loss term $\lambda\|w - \hat{w} _ 1\|^2$ anchors the model to the pretrained checkpoint, and (2) the LMC term $\beta \mathbb{E} _ {\alpha} \mathcal{L} _ 2(\hat{w} _ 2 + \alpha(w - \hat{w} _ 2))$  ensures smooth connectivity to the fine-tuned model. This creates a stable optimization landscape that naturally prevents excessive drift.

---

### Official Review · Reviewer_wfKc · 2025-10-31

**Soundness:** 4
**Presentation:** 3
**Contribution:** 3
**Rating:** 6
**Confidence:** 3

**Summary:**

The paper proposed a novel model merging method named Continue fine-tuning, which utilizes the low-loss path phenomenon in the model merging to effectively combine knowledge from the pre-training model and the fine-tuned version, achieving high performance across various tasks, models, and fits well with different tuning methods.

**Strengths:**

- The proposed framework shows a strong performance gain compared to the simple model merging baseline
- Achieving competitive performance with low inference cost
- Theoretical analysis on the low-loss path phenomenon and its relation with easing catastrophic forgetting is interesting.

**Weaknesses:**

- While overall performance gain looks good, the improvement on the different tuning methods is unstable, sometimes hurts the base performance while largely improves the novel class, and sometimes degrades the novel performance while the base performance becomes better. This indicates the instability of the proposed framework
- Figure 1 is really unfriendly for the reader. I strongly suggest that the author update the scale for different datasets to emphasize the performance difference between models instead of packing them together...

**Questions:**

- Does the low-loss path phenomenon always exist, no matter how complex the models are? Since the more recent VLM architecture has become increasingly complex, handling complex real-world tasks that generalize across various tasks, the loss space will become much more complicated. I am curious whether such LMCs discovered at an early age still affect?
- How is Figure 2 constructed? Is it just an illustration or sampled from the training process? Could the author provide more detail on the illustration part.

I will raise my score once my concerns are solved.

---

> ### Author Response · Authors · 2025-11-19
>
> **W1**: Stability of the proposed framework.
>
> **R**: Thanks for this point. The varying patterns of base vs. novel performance improvements across datasets reflect expected behaviour rather than instability.
>
> Evaluated across 11 datasets in Table 1, all base methods achieve improvements in both base and novel accuracy when combined with our MergeTune, except CoOp+MergeTune, which shows slightly lower base accuracy than CoOp (80.82 vs. 82.69). This is due to CoOp's extreme overfitting to base classes; however, our method achieves substantial novel class gain (+10.75%) and an overall harmonic mean increase (+5.58%). It is worth noting that the most critical metric is the harmonic mean, which accounts for both the base and novel performance tradeoffs. Our method achieves consistent harmonic mean improvements across all baselines (CoOp: +5.58%, KgCoOp: +0.97%, MMA: +0.57%, PromptKD: +0.36%), with larger gains for methods suffering from severe catastrophic forgetting.
>
>
> **W2**: Readability of Figure 1.
>
> **R**: Thank you for this suggestion. We have revised the Figure1 in the updated PDF by using independent scales for each dataset axis, making the differences between methods more visually apparent and reader-friendly as you recommended. Please let us know if further refinement is suggested.
>
>
> **Q1**: Generalisation to recent VLM architecture?
>
> **R**: Thank you for this suggestion. To demonstrate MergeTune's scalability, we have now conducted additional experiments on larger vision-language models CLIP ViT-L/14, and with a different VLM family SigLIP [1]. **Table 6** in the updated PDF presents comprehensive base-to-novel generalisation results across three model architectures for all 11 datasets. The following summary shows averaged performance over 11 datasets:
>
> - **CLIP-L/14** (CLIP large): CoOp + MergeTune achieves 82.14% HM (+1.93% over CoOp baseline 80.21%)
> - **SigLIP2-B/16** (a different VLM backbone): CoOp + MergeTune achieves 80.26% HM (+1.60% over CoOp baseline 78.66%)
> - **SigLIP2-L/16** (SigLIP large): CoOp + MergeTune achieves 83.73% HM (+1.30% over CoOp baseline 82.43%)
>
> These results demonstrate that **MergeTune consistently improves performance across different model scales and architectures**, validating its generalizability beyond CLIP ViT-B/16 and CLIP.
>
> [1] Tschannen M, Gritsenko A, Wang X, et al. Siglip 2: Multilingual vision-language encoders with improved semantic understanding, localization, and dense features. arXiv:2502.14786, 2025.
>
>
> **Q2**: How is Figure 2 constructed?
>
> **R**: Sorry for the confusion. Figure 2 is a conceptual illustration designed to convey the core idea of our method, not actually sampled from the training process. We chose illustration over real training visualisation because the loss landscape for the used VLMs is high-dimensional and cannot be directly visualised in 2D/3D space without dimensionality reduction, which requires projection methods (e.g., PCA) that might obscure the key concepts.
>
> To provide empirical validation of the low-loss connectivity illustrated in Figure 2, we have now **added Figure 4** in the supplementary, which shows the results of interpolated weights between our continued model and both endpoints (zero-shot CLIP and fine-tuned model). These curves confirm that MergeTune successfully establishes the smooth, low-loss paths depicted conceptually in Figure 2. We have now revised the caption to clarify that Figure 2 is a schematic illustration.

---

### Comment · Area_Chair_s9TC · 2025-11-23
**The authors' rebuttal is available. Please read, comment, and discuss.**

Dear Reviewers,

Thanks for your time and effort in reviewing ICLR2026 submissions. The authors have provided their responses to your review. Please read and raise your further comments, and discuss with the authors.

Best regards,

Your AC

---

### Author Response · Authors · 2025-12-02
**Final Summary**

Dear AC,

Thank you for taking the time to review our submission. We have included a summary below to hopefully ease the subsequent review process.

In short, our overall ratings improved from [4, 6, 6, 6] to [6, 6, 6, 6] on Nov. 25. You can see the change in the discussion. (Reviewer Poyv explicitly confirmed that all concerns were resolved and raised the score from 4 to 6 on Nov. 25)

More specifically, during the rebuttal, we mainly responded to the following concerns and updated some parts of our paper:

**Generalizability and scalability to other VLM architectures**

We conducted new experiments on a larger CLIP backbone (CLIP-L/14) and a different VLM family (SigLIP2 [1]), demonstrating consistent generalisation across different model scales and architectures.

**Theoretical assumptions**

• Gradient = zero for the pretrained checkpoint

We justified this assumption by citing prior ICLR work (BRECQ [2], SQuant [3]) and empirically confirmed a small L2 gradient norm on an ImageNet-1k proxy, demonstrating that our approximation is practically valid.

•  Isotropic Hessian approximation

We validated the isotropic curvature assumption by comparing it with a Fisher Information Matrix (anisotropic surrogate), showing nearly identical performance and demonstrating robustness to this curvature assumption.

**Computational cost**

We quantified the training overhead (around 3-5x the base VLM finetuning, depending on the number of interpolation samples) and clarified that this translates to only 2-3 additional GPU hours on ImageNet. This extra cost is negligible compared to the massive computational cost of VLM pretraining, yet it consistently and effectively restores the VLM’s pretrained knowledge.

Furthermore, unlike ensemble methods, MergeTune produces a single model, resulting in no test-time inference overhead and effectively compensating for the additional training cost.

**Paper update**

We revised Fig. 1 with dataset-specific scales for better readability. We added a visualisation of weight interpolation (Fig. 4 in the supplementary), empirically confirming the smooth, low-loss paths after our continued finetuning, and other analyses, including α-sampling behaviour (saturation at $N_\alpha≥10$) and whether over-merging occurs.

**References**

[1] Tschannen M, Gritsenko A, Wang X, et al. Siglip 2: Multilingual vision-language encoders with improved semantic understanding, localization, and dense features. arXiv:2502.14786, 2025.

[2] Li Y, Gong R, Tan X, et al. BRECQ: Pushing the limit of post-training quantization by block reconstruction. ICLR 2021.

[3] Guo C, Qiu Y, Leng J, et al. SQuant: On-the-fly data-free quantization via diagonal hessian approximation. ICLR 2022.

Sincerely,

The Authors

---

### Meta-Review · Area_Chair_D63a · 2025-12-26

**Summary:**

The reviewers raised a series of core concerns that inform the final decision, including the generalization and scalability of MergeTune to larger VLMs and non-CLIP model families, the theoretical justification for the key assumptions adopted in the method, the computational overhead introduced by the proposed approach, the stability and consistency of the reported performance gains, and the sufficiency of empirical validations for the method’s design choices.

**Reviewer Concerns:**

All key reviewer concerns were addressed by the rebuttal:
- Generalization: Authors added experiments on CLIP-L/14 and SigLIP variants to validate scalability.
- Theoretical assumptions: Empirical tests (gradient norm on ImageNet-1k, Fisher Information Matrix comparison) and literature citations justified the surrogate loss.
- Computational cost: The authors contextualized training overhead (minimal GPU hours vs. VLM pretraining) and noted no test-time inference cost.
- Performance stability: Supplementary analyses (feature similarity, training duration) confirmed consistency and ruled out "over-merging."

No concerns remain outstanding.

**Reviewer Scores:**

- Reviewer wfKC: Initially scored 6 (marginal acceptance). Their concerns (stability, figure readability, generalization) were fully addressed; their score would likely remain 6 or increase slightly.
- Reviewer wRZ r: Initially scored 6 (marginal acceptance). All their concerns (theoretical justification, computational cost, scalability) were resolved; their score would remain 6.
- Reviewer uAR: Initially scored 6 (marginal acceptance). Their concerns (assumption validation, generalization, hyperparameter selection) were addressed; their score would remain 6.
- Reviewer Poyv: Initially scored 4 (below threshold). After resolving their concerns, they explicitly raised their score to 6 (marginal acceptance).

---

### Decision · Program_Chairs · 2026-01-26

Accept (Poster)